# RTG: Reverse Trajectory Generation for Learning Rigid-Body Manipulation Under Sparse Reward

## Abstract

Deep Reinforcement Learning (DRL) under sparse reward conditions remains a long-standing challenge in robotic learning. In such settings, extensive exploration is often required before meaningful reward signals can guide the propagation of state-value functions. Prior approaches typically rely on offline demonstration data or carefully crafted curriculum learning strategies to improve exploration efficiency. In contrast, we propose a novel method tailored to rigid body manipulation tasks that addresses sparse reward without the need for guidance data or curriculum design. Leveraging recent advances in differentiable rigid body dynamics and trajectory optimization, we introduce the Reverse Rigid-Body Simulator (RRBS), a system capable of generating simulation trajectories that terminate at a user-specified goal configuration. This reverse simulation is formulated as a trajectory optimization problem constrained by differentiable physical dynamics. RRBS enables the generation of physically plausible trajectories with known goal states, providing informative guidance for conventional RL in sparse reward environments. Leveraging this, we present Reverse Trajectory Generation (RTG), a method that integrates RRBS with a beam search algorithm to produce reverse trajectories, which augment the replay buffer of off-policy RL algorithms like DDQN to solve the sparse reward problem. We evaluate RTG across various rigid body manipulation tasks, including sorting, gathering, and articulated object manipulation. Experiments show that RTG significantly outperforms vanilla DRL and improved sampling strategies like Hindsight Experience Replay (HER) and Reverse Curriculum Generation (RCG). Specifically, RTG is the only method that can solve each task with success rates of over 70% within given compute budget.

## 1 Introduction

Deep Reinforcement Learning (DRL) serves as the foundation for robot skill acquisition, enabling robots to learn and refine skills that optimize user-defined reward functions. When combined with expressive deep neural policies, RL has demonstrated remarkable success across various domains, including game-playing (Mnih et al., 2015), language-based reasoning (Havrilla et al., 2024), robot locomotion (Duan et al., 2016), and robotic manipulation (Yu et al., 2020; Mahmood et al., 2018). Despite these advances, sample efficiency remains a critical challenge, significantly limiting DRL's applicability in real-world scenarios, particularly in computationally constrained environments. This inefficiency primarily stems from several fundamental limitations of current DRL methodologies. First, general-purpose DRL relies heavily on exploration strategies to discover and connect useful state-transition samples through interaction with the environment. Though various efficient exploration techniques have been developed over the years (Ladosz et al., 2022), these methods often struggle to scale in complex settings. In practical scenarios, exploratory behaviors guided by general-purpose strategies can rapidly become intractable, making it difficult to consistently collect informative samples that yield high reward signals. Second, the formulation of reward functions further exacerbates inefficiency (Eschmann, 2021). In robotic tasks like tabletop manipulation, the most natural and straightforward reward design is tied to the (partial) completion of the task. However, such signals are often sparse and the majority of state-transition samples yield zero reward.

Several lines of research have sought to practically improve sample efficiency in DRL. The most widely adopted approach is off-policy learning (Mnih et al., 2015; Lillicrap et al., 2015), where previously collected transition data are repeatedly reused instead of discarding them after each policy update. This strategy significantly reduces the need for fresh interactions with the environment. However, the effectiveness of off-policy methods remains fundamentally constrained by the quality and diversity of the data. To address this limitation, researchers have turned to leveraging domain knowledge to bootstrap learning. One prominent line of work integrates expert demonstrations into the training pipeline (Rengarajan et al., 2022), enabling the agent to initialize policies or guide exploration with trajectories that encode meaningful behavior. Similarly, reward shaping techniques (Ng et al., 1999) incorporate task-specific prior knowledge into the reward function, effectively providing denser feedback and reducing the burden of pure trial-and-error learning. Nevertheless, acquiring high-quality demonstrations or carefully engineered shaping functions is often costly and impractical in real-world applications, where domain knowledge may be limited or noisy. In parallel, another promising direction focuses on improving sampling strategies by biasing the agent toward more informative experiences. Notable examples include Hindsight Experience Replay (HER) (Andrychowicz et al., 2017), which relabels unsuccessful trajectories with alternative goals to extract positive learning signals, and Reverse Curriculum Generation (RCG) (Florensa et al., 2017), which gradually increases task difficulty, starting from states close to successful completion.

Drawing on insights from HER and RCG, we propose a more general and efficient trajectory sampling strategy tailored to rigid body manipulation tasks. While HER has proven effective, its applicability is largely limited to single-object manipulation scenarios, where the object's goal state can be relabeled by translating it closer to the robot's current state to generate additional reward. However, this mechanism breaks down in more complex multi-object settings. For instance, consider a common robotic task where two objects must be pushed together on a table (Wang et al., 2023) as illustrated in Figure 1. In such cases, HER fails because intermediate states yield zero reward under arbitrary goal relabeling. By contrast, RCG offers

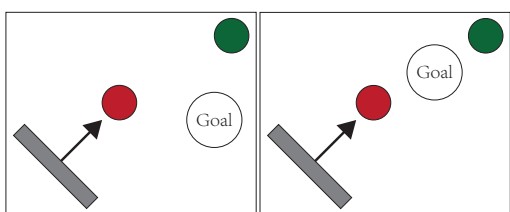

Figure 1: We consider the task of pushing two circular objects (red) and (green) together into the goal region (left), where the reward is non-zero when both objects are inside the goal region. HER works by moving the goal position to acquire non-zero reward. But the reward can be zero for any goal position at an intermediary state (right).

greater generality and can handle a wider range of manipulation tasks under sparse reward, provided that a suitable proximity metric for task completion is available. This metric enables the construction of progressively challenging curricula, guiding the agent from easier to harder tasks. Despite its generality, RCG still depends on general-purpose exploration within each stage of the curriculum.

We observe that the efficiency of HER and the generality of RCG can be unified by generating state samples in a reversed manner. Specifically, instead of relying solely on the forward state transition function $\mathcal{T}(s_{t+1} \mid s_t, a_t)$, we assume the transition function is time-invertible and introduce the reverse transition function $\bar{\mathcal{T}}(s_t \mid s_{t+1}, a_t)$, which predicts the preceding state $s_t$ given the next state $s_{t+1}$ and the control signal $a_t$. For the class of robotic manipulation tasks, the forward and reverse transition functions naturally correspond to forward and reverse physics simulations. Leveraging recent advances in optimization-based physics simulation (Gast et al., 2015) and differentiable contact mechanics (Huang et al., 2024), we show that sampling from both $\mathcal{T}(s_{t+1} \mid s_t, a_t)$ and $\bar{\mathcal{T}}(s_t \mid s_{t+1}, a_t)$ can be formulated as trajectory optimization problems in forward and reversed time, respectively. These problems can be solved efficiently using Sequential Quadratic Programming (SQP). Equipped with the ability to sample from $\bar{\mathcal{T}}(s_t \mid s_{t+1}, a_t)$, we propose the Reverse Trajectory Generation (RTG) algorithm, which actively generates trajectories starting from task-completed states with positive reward, utilizing a reverse rigid-body simulator (RRBS). RTG constructs reverse state-transition tuples by trajectory-optimizing control signals in RRBS. We integrate RTG-generated samples into off-policy DRL frameworks like DQN (Mnih et al., 2015) and DDQN (Van Hasselt et al., 2016) and evaluate the method across various rigid body manipulation tasks. Experiments on tabletop rigid-body manipulation demonstrate that RTG significantly outperforms vanilla DRL, simplified-task HER, and RCG.

## 2 RELATED WORK

**Sparse Reward** in RL is commonly encountered in robotic manipulation tasks, where the reward signal typically reflects task completion. In such settings, problem-agnostic DRL often struggles to explore the sparse, high-reward regions of the state space. To address this, several methods leverage domain knowledge in various forms. For instance, reward shaping (Ng et al., 1999) has been applied to provide dense, distance-based guidance (Trott et al., 2019) or to promote self-exploration (Bellemare et al., 2016). Additionally, expert demonstrations can bootstrap the learning process through inverse RL (Abbeel & Ng, 2004) and imitation learning (Ross et al., 2011; Ho & Ermon, 2016). However, these approaches often rely on high-quality domain knowledge, which must be specifically tailored to each robotic manipulation task. To reduce the need for manual specification, recent works (Zheng et al., 2018; Memarian et al., 2021; Devidze et al., 2022) propose self-supervised tuning of the reward signal through bilevel optimization, adjusting the parameters of the reshaped reward function to maximize the return under the original sparse reward.

**HER and RCG** are two complementary strategies designed to bias trajectory sampling in order to increase the likelihood of visiting high-reward regions. HER (Andrychowicz et al., 2017) is applied in goal-conditioned tasks, where the task can be relabeled to bring the goal closer to the current state, yielding higher reward. Since its inception, HER has been successfully generalized to a variety of settings, including dynamic goal configurations (Packer et al., 2021), visual domains (Sahni et al., 2019), and meta-RL (Packer et al., 2021), among others. Despite its versatility, HER remains limited in application to tasks where the goal can be relabeled to generate high reward, which is not always feasible in more complex multi-object manipulation tasks, as illustrated in Figure 1. In contrast, RCG (Florensa et al., 2017) focuses on starting from initial states that are closer to high-reward regions, effectively reducing the distance to the goal. However, RCG's effectiveness hinges on the careful design of curricula, which in turn requires domain-specific knowledge, including reversible dynamics (Florensa et al., 2017), approximate distance functions, state demonstrations (Resnick et al., 2018), and guiding policies (Uchendu et al., 2023). Additionally, RCG's performance is constrained by the exploration capabilities of downstream DRL methods.

**Physics Simulators** play a crucial role in downstream robot learning and manipulation tasks. While rigid body simulators have matured significantly (Erez et al., 2015), consistently delivering high-fidelity trajectory data with impressive performance. One avenue of research aims to improve simulation performance by leveraging massively parallel processors (Xu et al., 2022). However, we argue that such improvements alone do not address the challenge of sparse reward, as the state space can grow exponentially more complex than the performance gains achieved by optimized simulators. Another promising direction involves incorporating physics simulators into model-based DRL. This can be done by deriving analytic policy gradients through backpropagation (Son et al., 2023), or by using differentiable physics simulations to maximize trajectory-wise returns via local optimization (Levine & Koltun, 2013; Mordatch & Todorov, 2014). In contrast, by exploiting the forward-reverse differentiability of state-of-the-art optimization-based simulators (Huang et al., 2024), we demonstrate that simulations can be performed in reversed time by solving trajectory optimization problems.

**Multi-Object Manipulation** is a common challenge in robot learning, which has received increasing attention and been extensively reviewed in Pan et al. (2022). Multi-object manipulation, particularly under sparse reward conditions, presents a significant challenge for DRL, and a variety of domain knowledge has been explored to enable successful policy learning. For instance, differentiable dynamics (Wan et al., 2024) and learned dynamics (Li et al., 2020b) have been employed to synthesize control through trajectory optimization. Structured policy parameterization (Li et al., 2020a; Haramati et al., 2024) has been used to transfer learned skills from simpler tasks to more complex ones, such as in the case of RCG. Additionally, reward shaping and Monte Carlo Tree Search (MCTS) have been combined to effectively search for multi-object sorting policies (Song et al., 2020). In contrast, our RTG method does not rely on any additional domain knowledge and can be seamlessly integrated with these techniques to further enhance performance.

## 3 PRELIMINARIES: DRL UNDER RIGID BODY DYNAMICS

We consider the standard DRL setting, where an agent interacts with an environment governed by rigid body dynamics. We assume the environment is fully observable. Formally, the environment is

defined by a state space $\mathcal{S}$, an action space $\mathcal{A}$, and an initial state distribution $p(s_0)$, where we only consider discrete action space in this work. A deterministic policy $\pi$ maps a state $s_t \in \mathcal{S}$ to an action $a_t \in \mathcal{A}$. At the beginning of each episode, the agent samples an initial state $s_0 \sim p(s_0)$. At every timestep $t$, the agent selects an action according to $a_t = \pi(s_t)$ and receives a reward $r_t = r(s_t, a_t)$, where $r$ is the reward function. The environment then transitions to the next state according to the transition distribution $s_{t+1} \sim \mathcal{T}(\bullet \mid s_t, a_t)$. The objective of DRL is to maximize the expected cumulative return, defined as $\mathbb{E}_{s_0 \sim p}\left[\sum_{t=0}^{\infty} \gamma^t r_t \mid s_0\right]$, where $\gamma \in (0, 1)$ is the discount factor.

**Off-Policy DRL** In this work, we adopt the standard off-policy Deep Q-Learning (DQN) framework (Mnih et al., 2015). In sparse reward settings, states outside the goal region $\mathcal{G}$ consistently yield zero reward, which presents a major challenge for DRL in discovering high-reward regions. To learn a near-optimal policy, DQN approximates the optimal state-action value function $Q(s, a) = \mathbb{E}\left[\sum_{t=0}^{\infty} \gamma^t r_t \mid s_0 = s, a_0 = a\right]$ using a neural network $Q_\theta(s, a)$. The optimal value function satisfies the Bellman equation: $Q(s_t, a_t) = r_t + \max_a Q(s_{t+1}, a)$. Accordingly, DQN trains $Q_\theta$ by minimizing the Bellman loss: $\mathcal{L} = \mathbb{E}_{\mathcal{D}}\left[(Q_\theta(s_t, a_t) - y_t)^2\right]$, where the target value is given by $y_t = r_t + \max_a Q_{\theta'}(s_{t+1}, a)$ and $\theta'$ denotes the parameters of the target network. The loss is computed over a replay buffer $\mathcal{D} = \{(s_t, a_t, r_t, s_{t+1})\}$ containing transition tuples collected through interaction with the environment. The quality of these transition tuples is critical for the sample efficiency of DQN. However, in sparse reward settings, where $r(s_t, a_t) = \mathbb{1}[s_t \in \mathcal{G}]$ and $\mathcal{G}$ represents a small subset of the state space corresponding to successful task completion, to obtain such high-quality samples is particularly challenging.

**Rigid Body Dynamics** We focus on robot manipulation tasks where the environment is composed entirely of rigid bodies, which is a standard setting in robot learning that encompasses a wide range of manipulation scenarios. In this context, the transition function $\mathcal{T}(s_{t+1} \mid s_t, a_t)$ is governed by a deterministic rigid body simulator. Mature simulation algorithms like Erez et al. (2015) can produce highly accurate trajectory data with excellent performance. Recent advances have further improved these simulators by introducing fully differentiable structures, primarily through optimization-based approaches (Huang et al., 2024; Romanyà-Serrasolsas et al., 2025). These methods formulate the transition function as a deterministic implicit function $\Lambda(s_{t+1} \mid s_t, c_t) = 0$, making it differentiable with respect to all three variables, where $c_t$ is the continuous control signal, such as joint torques and forces on the robot. Forward simulation is then performed by solving for $s_{t+1} = \Lambda^{-1}(s_t, c_t)$ using the inverse function theorem, which can be practically computed via Newton's method. In addition, the differentiable structure allows us to compute the state- and action-derivatives $ds_{t+1}/ds_t$ and $ds_{t+1}/dc_t$. Such a differentiable structure has been leveraged in prior works like Son et al. (2023); Levine & Koltun (2013); Mordatch & Todorov (2014) to improve learning stability and efficiency.

# 4 DRL WITH REVERSE TRAJECTORY GENERATION (RTG)

We propose RTG, a method that combines the sample efficiency of HER with the generality of RCG, further improving the performance of DRL under sparse reward. Our key observation is that conventional DRL methods sample trajectories $(s_0, s_1, \cdots, s_T)$ through forward simulations, starting from an initial state distribution that is typically far from the goal region $\mathcal{G}$. Due to the high variance in future state distributions, it becomes increasingly unlikely for the final state $s_T$ to lie within $\mathcal{G}$. Even when using trajectory optimization techniques with differentiable simulations (Levine & Koltun, 2013; Xing et al., 2024), the probability of reaching high-reward regions remains low. This is because sparse reward functions yield zero gradients outside the goal region, making optimization ineffective. To address this, we propose the assumption that the state transition function is invertible and sample trajectories in a time-reversed manner, conceptually corresponding to sampling from the distribution $\tilde{\mathcal{T}}$. By starting from a known goal state $s_T \in \mathcal{G}$—where task completion is guaranteed— we ensure exploration of high-reward regions, thereby significantly improving sample efficiency in off-policy DRL. If sampling from $\tilde{\mathcal{T}}$ is made tractable, RTG inherits the strengths of both HER and RCG. Compared to HER, RTG generalizes beyond single-object to multi-object manipulation tasks, as it only requires that states within the goal region $\mathcal{G}$ can be sampled. For instance, in the toy example illustrated in Figure 1, we can randomize the positions of two circles within a randomly positioned goal region. In contrast to RCG, RTG applies the reverse sampling concept at the trajectory level rather than the curriculum level. As a result, RTG does not depend on the underlying DRL agent's exploration capabilities.

## 4.1 RRBS with Time Reversed Trajectory Sampling

In this section, we present the Reverse Rigid-Body Simulator (RRBS), a system capable of generating simulation trajectories that terminate at a user-specified goal configuration. Generally speaking, it is challenging to sample from $\tilde{\mathcal{T}}$ for an arbitrary forward transition function $\mathcal{T}$. In the stochastic setting, it is well-known that deriving $\tilde{\mathcal{T}}$ via Bayes' rule is intractable (Kingma & Welling, 2014). Fortunately, we show that for optimization-based physics simulators (Gast et al., 2015; Huang et al., 2024) with fully differentiable structures, it is possible to approximately sample from $\tilde{\mathcal{T}}$ by solving trajectory optimization problems. Specifically, suppose the simulator is defined by a fully differentiable implicit function $\Lambda(s_{t+1}, s_t, c_t, q(a_t)) = 0$. Here we condition the implicit function on an additional term $q(a_t)$, which is denoted as the discrete action-dependent configuration. In this work, we consider robot manipulation tasks with discrete action space. For example, in a robot pushing task, the robot can choose the pushing position and orientation by selecting action $a_t$. In this case, we can model $q(a_t)$ as the position and orientation of the robot end-effector.

Under our setup, forward simulation can then be performed by using the Newton's method to solve: $\arg\min_{s_{t+1}} \|\Lambda(s_{t+1}, s_t, c_t, q(a_t))\|^2$. Similarly, given $s_{t+1}$, $a_t$, and $c_t$, we can perform time-reversed simulation by solving for $s_t$ via:

$$\arg\min_{s_t} \|\Lambda(s_{t+1}, s_t, c_t, q(a_t))\|^2. \tag{1}$$

However, we argue that this approach is impractical for DRL training. The optimal action $a_t = \pi(s_t)$ depends on the previous state $s_t$, which is unknown when starting from $s_{t+1}$. As a result, we cannot determine the corresponding optimal action $a_t$, making the reverse simulation ill-posed. This issue has also been noted in Barkley et al. (2024). Further, the continuous control signal $c_t$ is unknown a priori. For example, it is non-trivial to infer the robot joint torques and forces in order for the robot to push an object along a given direction. Instead, we propose the following physics-constrained optimization, which allows us to search for the continuous control signal $c_t$ that reaches a user-specified previous state $s_t$:

$$\arg\min_{s_t, c_t} \mathcal{O}(s_t, a_t) + \lambda \|c_t\|^2 \quad \text{s.t.} \quad \Lambda(s_{t+1}, s_t, c_t, q(a_t)) = 0, \tag{2}$$

where $\mathcal{O}$ is a user-defined objective function, and our second term serves as a minimal-effort regularization weighted by $\lambda$ to stabilize the optimization. Compared to Equation 1, this formulation offers two key advantages. First, it enables automatic determination of the control signal $c_t$. Second, it introduces a flexible, state-dependent objective function $\mathcal{O}$, which can be easily defined based on the robot action specification. For example, if our action $a_t$ requires the robot to push an object in the direction of $d(a_t)$, then we can define the objective function as:

$$\mathcal{O}(s_t, a_t) = \left(x_t^i, y_t^i\right) d(a_t), \tag{3}$$

where $\left(x_t^i, y_t^i\right)$ is the position of the object to be pushed. That is, we maximize the pushing distance along the negative pushing direction to reflect the time-reversed nature of reverse simulation.

Finally, we propose an $h$-step generalization of Equation 2, *i.e.*, we jointly optimize over a sequence of $h$ consecutive states:

$$\begin{aligned} \arg\min_{\substack{s_t, \ldots, s_{t-h+1} \\ c_t, \ldots, c_{t-h+1}}} \quad & \mathcal{O}(s_{t-h+1}, a_t) + \lambda \sum_{k=0}^{h-1} \|c_{t-k}\|^2 \\ \text{s.t.} \quad & \Lambda(s_{t-k+1}, s_{t-k}, c_{t-k}, q(a_t)) = 0 \quad \forall k = 0, \cdots, h-1, \end{aligned} \tag{4}$$

which effectively performs physics-constrained trajectory optimization over a time horizon of $h$ steps. This $h$-step formulation is particularly useful in robotic manipulation tasks, where actions such as pushing or sliding typically span multiple timesteps during which the robot applies the same action, *e.g.*, pushing direction and distance. Equation 4 allows us to optimize the full state trajectory over the duration of an entire robot action. We denote Equation 4 as the action-dependent reverse sampling function $(s_{t-h+1}, \cdots, s_t) = \text{RS}^h(s_{t+1}, a_t)$. We refer readers to Appendix A for more details of this algorithm.

Note that solving time-reversed simulations is significantly slower than solving forward simulations. This is because forward simulation involves $h$ decoupled Newton solves, each independent of the others. In contrast, the reverse simulation couples $h$ consecutive states into a single trajectory optimization problem, as formulated in Equation 4. Nevertheless, we show that this optimization can be solved efficiently using Sequential Quadratic Programming (SQP) (Boggs & Tolle, 1995). Thanks to the sparse dependencies between consecutive states, we can exploit the tridiagonal sparsity pattern of the Hessian matrix to accelerate the underlying linear solve (Jordana et al., 2025). As proven in Appendix A.3, , the per-iteration computational cost of our SQP remains linear in the trajectory length, *i.e.*, $O(h)$.

## 4.2 FORWARD REPLAY

One challenge with trajectory optimization is that numerical solvers rarely converge to exactly physically consistent solutions, *i.e.*, $\Lambda(s_{t-k+1}, s_{t-k}, c_{t-k}, q(a_t)) = 0$, due to numerical errors. This mismatch introduces discrepancies between forward and reverse simulations. To mitigate this issue, we adapt a forward replay procedure. After trajectory optimization, we run the forward simulator for each $k = 1, \ldots, h$ using the optimized control signal $c_{t-k}$, updating states as $s_{t-k+1} \leftarrow \arg\min_{s_{t-k+1}} \|\Lambda(s_{t-k+1}, s_{t-k}, c_{t-k}, q(a_t))\|^2$. In our experiments, forward replay effectively reduces distributional bias and accelerates convergence.

## 4.3 RTG WITH EXPLORATION VIA BEAM SEARCH

In this section, we demonstrate how the reversed simulator RRBS can be leveraged to guide DRL toward high-reward regions. A naïve strategy is to rely exclusively on the reverse simulator to generate state-transition tuples, from which the policy can be trained directly via the Bellman loss. While this approach is highly data-efficient, it is often computationally prohibitive in practice, since generating samples with the reverse simulator requires solving a large number of trajectory optimization problems via Equation 4. Furthermore, because the reverse simulator cannot exploit the learned policy $\pi$ to propose actions, it lacks the ability to balance state exploration with policy exploitation. To address these limitations, we draw inspiration from off-policy reinforcement learning with offline data, in particular Reinforcement Learning with Prior Data (RLPD) (Ball et al., 2023). We treat reverse-sampled trajectories as additional expert demonstrations, denoted by $\tilde{\mathcal{D}}$. These offline trajectories generated by RTG can be seamlessly integrated into any off-policy RL algorithm, in the same spirit as RLPD. Concretely, we adopt the symmetric sampling strategy proposed in RLPD: for each training batch, 50% of samples are drawn from the online replay buffer $\mathcal{D}$, and the remaining 50% from the reverse dataset $\tilde{\mathcal{D}}$, following the scheme of Ross & Bagnell (2012). This design is also consistent with the recent findings of Tao et al. (2024), which show that resetting agents to more difficult-to-reach states improves sample efficiency of DRL. In our framework, we accordingly initialize the agent at a reset probability of 50% for offline visited state distribution and 50% for task initial state distribution $p(s_0)$.

The remaining challenge in our algorithm design is to determine how to generate time-reversed trajectories via the trajectory optimization in Equation 4 in order to populate $\tilde{\mathcal{D}}$. To this end, we leverage the initial state distribution $p(s_0)$. Specifically, we randomly sample a pair consisting of a goal state $s_T \in \mathcal{G}$ and a candidate initial state $s_0^\star \sim p(s_0)$, where $s_0^\star$ serves as the target initial state. Analogous to forward exploration in off-policy DRL, we perform time-reversed exploration: starting from $s_T$, we recursively call the $\mathrm{BS}^h$ function to generate $h$ preceding states under different actions $a \in \mathcal{A}$. For the discrete action space, this procedure expands into a tree of length-$h$ sub-trajectories with branching factor $|\mathcal{A}|$, where we denote each node on the tree as $n = (s_{t-h+1}, \cdots, s_{t+1}, a_t)$. For complex manipulation tasks, however, the resulting tree quickly becomes intractably large. To balance computational feasibility with sufficient state-space coverage, we adopt a beam search strategy (Tillmann & Ney, 2003). At each tree depth, only the top-$B$ most promising nodes are retained, ranked by their closeness to the target initial state $s_0^\star$ as measured by the ranking function $r(s_t) = \|s_t - s_0^\star\|_M$ with $M$ being a mask matrix detailed in Section 5.2. The process terminates when the tree reaches a predefined depth or when improvements in the ranking function fall below a threshold. The overall beam search procedure is summarized in Algorithm 1.

---

**Algorithm 1** Beam-Search($s_0^\star, s_T, B, d_{\max}, \delta r$)

---

1: **function** SELECT-TOP-NODES($\mathcal{S}_{\text{candidate}}, B$)               ▷ Select top $B$ nodes by closeness to $s_0^\star$
2:     **for** each $s_t \in \mathcal{S}_{\text{candidate}}$ **do**
3:         Compute rank $r(s_t) \leftarrow \|s_t - s_0^\star\|_M$
4:     Return top $B$ nodes $n \in \mathcal{S}_{\text{candidate}}$ with smallest $r(s_t)$
5: ▷ Beam search with limited breadth
6: $r_{\text{best}} \leftarrow \infty, \bar{\mathcal{D}} \leftarrow \varnothing, \mathcal{S}_{\text{active}} \leftarrow \{s_T^\star\}$
7: **for** $d = 1, \cdots, d_{\max}$ **do**
8:     $\mathcal{S}_{\text{candidate}} \leftarrow \varnothing$                                    ▷ Generate candidate set
9:     **for** each node $s_t \in \mathcal{S}_{\text{active}}$ **do**
10:        **for** each action $a \in \mathcal{A}$ **do**
11:            $(s_{t-h+1}, \cdots, s_t) = \text{RS}^h(s_{t+1}, a)$ Equation 4       ▷ Generate new node via TrajOpt.
12:            $\bar{\mathcal{D}} \leftarrow \bar{\mathcal{D}} \cup \{(s_{t-h+1}, a, r(s_{t-h+1}, a), s_{t+1})\}$   ▷ Populate state-transition dataset $\bar{\mathcal{D}}$
13:            $\mathcal{S}_{\text{candidate}} \leftarrow \mathcal{S}_{\text{candidate}} \cup \{s_{t-h+1}\}$
14:     $\mathcal{S}_{\text{active}} \leftarrow$ Select-Top-Nodes($\mathcal{S}_{\text{candidate}}, B$)           ▷ Select top-$B$ candidate nodes
15:     $h_{\text{best}}' \leftarrow \min\{r(s_t) : s_t \in \mathcal{S}_{\text{active}}\}$
16:     **if** $h_{\text{best}}' > h_{\text{best}} - \delta r$ **then**
17:         Break
18:     $h_{\text{best}}' \leftarrow h_{\text{best}}$
19: Return $\bar{\mathcal{D}}$

---

## 5 EVALUATION

We consider a multi-object table-top manipulation problem similar to Huang et al. (2019), where there are $N$ rigid objects on the table. To model rigid body physics, we implement the 2D rigid body simulator using the formulation proposed in Huang et al. (2024). In this case, the state $s_t$ is a concatenation of $M$ rigid body dynamic configurations, denoted as $s_t = (s_t^1, s_t^2, \cdots, s_t^M)$ where each $s_t^i = (x_t^i, y_t^i, \phi_t^i, \dot{x}_t^i, \dot{y}_t^i, \dot{\phi}_t^i)$ with the first 3 elements being the position and orientation on the 2D table-top and the last 3 elements being the corresponding velocity.

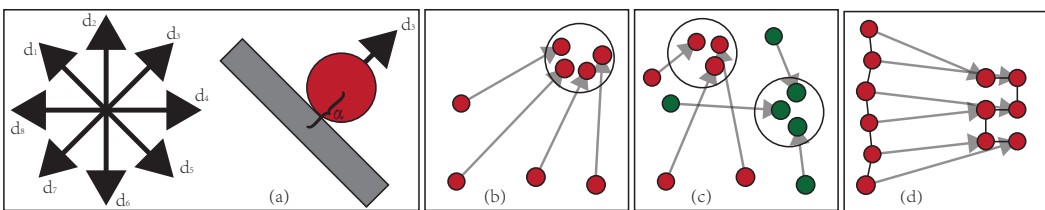

Figure 2: (a): Our action requires the robot to push an object along $N = 8$ uniformly spaced directions for a fixed distance. The initial pushing position is the computed by finding the largest negative $\alpha$ with no collision between objects and robot end-effetor. (b): The task of Gathering requires all objects to fall inside the goal region. (c): The task of Sorting requires two types of objects to fall into designated region as determined by their labels. (d): The task of ArtManip. requires pushing a hinge-connected articulated body into an S-shaped target pose.

### 5.1 RIGID BODY MANIPULATION TASKS

Following the setting in Li et al. (2018), we only allow the robot to manipulate objects via non-prehensile pushing using a set of discrete pushing actions. As illustrated in Figure 2 (a), the pushing direction is discretized into $N$ uniformly spaced directions on the plane; without loss of generality, we set $N = 8$ in our experiments, *i.e.*, 8 equispaced directions $(d^1, \cdots, d^8)$ over a fixed distance. For each direction $d^j$ and each object $s_t^i$, we define the initial pushing position of the robot end-effector as $d^i \alpha + (x_t^i, y_t^j)$, where we choose $\alpha$ as the largest negative value such that the robot end-effector does not intersect any object. As such, the action space $\mathcal{A}$ is discretized into $N \times M$ manipulation primitives, *i.e.*, $|\mathcal{A}| = 8M$. We evaluate our method on three manipulation tasks under absolute sparse reward, which are illustrated in Figure 2 (b-d) and listed below:

- **Gathering:** The goal is to gather all objects into a single designated target area $\mathcal{G}$, which is an circular area with radius $R$ centered at $\left(x^{\mathcal{G}}, y^{\mathcal{G}}\right)$ and we define the reward as $r(s_t, a_t) = \prod_{i=1}^{M} \mathbf{1}\left[\left(x_t^i, y_t^i\right) \in \mathcal{G}\right]$.

- **Sorting:** The goal of sorting is to divide objects into 2 clusters, with each object equipped with a label $l(i)$. Each cluster has a designated goal region $\mathcal{G}^1$ and $\mathcal{G}^2$, both with radius $R$ and centered at $\left(x^{\mathcal{G}^1}, y^{\mathcal{G}^1}\right)$ and $\left(x^{\mathcal{G}^2}, y^{\mathcal{G}^2}\right)$, respectively. We define the reward as: $r(s_t, a_t) = \prod_{i=1}^{M} \mathbf{1}\left[\left(\left(x_t^i, y_t^i\right) \in \mathcal{G}^1 \wedge l(i) = 1\right) \vee \left(\left(x_t^i, y_t^i\right) \in \mathcal{G}^2 \wedge l(i) = 2\right)\right]$.

- **Articulated Manipulation (ArtManip.):** In this case, we assume the $M$ objects are connected using hinge joints, each of which allows free rotation in the range $[-\pi, \pi]$. And the goal is for the robot to push objects so that the entire articulated body takes a given target pose, which is defined by a set of $M$ target positions $\left(x^{\mathcal{G}^i}, y^{\mathcal{G}^i}\right)$. We define the reward as: $r(s_t, a_t) = \prod_{i=1}^{N} \mathbf{1}\left[\|\left(x_t^i, y_t^i\right) - \left(x^{\mathcal{G}^i}, y^{\mathcal{G}^i}\right)\| \leq \epsilon\right]$. This task is more challenging as pushing one object may affect the poses of the others.

## 5.2 BASELINES

To demonstrate that our method can be combined with various off-policy DRL algorithms, we build our RTG framework with two variants, Deep Q-Networks (DQN) (Mnih et al., 2015) and Double DQN (DDQN) (Van Hasselt et al., 2016). We compare our method with two baselines: RCG and HER. The RCG assumes the availability of a distance-to-goal metric, for which we use our ranking function $r(s_t)$, where we define our mask matrix $M$ to only measure the distance between object center positions, ignoring orientations. Specifically, we generate the $i$-th curriculum with initial states satisfying $r(s_0) < \epsilon_i$. The comparison with HER is tricker as HER requires a goal conditioned and does not generalize to multi-object tasks as illustrated in Figure 1. Instead, we propose an implicit goal-conditioning setting. Take the gathering task for example, we use a designated goal position $\left(x^{\mathcal{G}}, y^{\mathcal{G}}\right)$. Now suppose we condition our task on this goal position, HER works by moving the goal position to maximize the reward. This is equivalent to using the modified reward function: $r_{\text{HER}}(s_t, a_t) = \max_{(x^{\mathcal{G}}, y^{\mathcal{G}})} r(s_t, a_t)$, denoted with subscript $_{\text{HER}}$. Note that using $r_{\text{HER}}$ yields a simpler task than the original HER formulation, since it obviates the requirement for Universal Value Function Approximators (UVFA) (Bellemare et al., 2016) and the reward function is engineered to maximize reward over all goals. We call such a reward function implicit HER or I-HER. Similarly, we can define I-HER reward for the sorting task by treating both $\left(x^{\mathcal{G}^1}, y^{\mathcal{G}^1}\right)$ and $\left(x^{\mathcal{G}^2}, y^{\mathcal{G}^2}\right)$ as goals and define: $r_{\text{HER}}(s_t, a_t) = \max_{\|(x^{\mathcal{G}^1}, y^{\mathcal{G}^1}) - (x^{\mathcal{G}^2}, y^{\mathcal{G}^2})\| \geq 2R} r(s_t, a_t)$, where we require the center of two circles to be larger than $2R$ to reflect the requirement of sorting. Finally, for ArtManip., we assume that the goal pose of the articulated body can undergo arbitrary rigid transformations and define:

$$r_{\text{HER}}(s_t, a_t) = \max_{\delta x, \delta y, \delta \theta} \prod_{i=1}^{N} \mathbf{1}\left[\|\left(x_t^i, y_t^i\right) - R(\delta\theta)\left(x^{\mathcal{G}^i}, y^{\mathcal{G}^i}\right) - (\delta x, \delta y)\| \leq \epsilon\right],$$

with $R(\delta\theta)$ denoting a 2D rotation matrix.

## 5.3 RESULTS

The main results are summarized in Figure 3, Figure 4, Figure 5, showing how our methods outperform all baselines across all three tasks. RTG with both DDQN and DQN are the only methods that

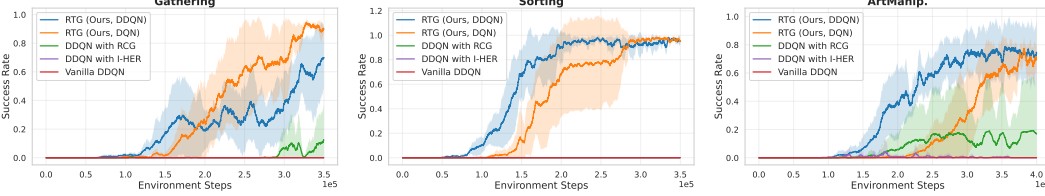

Figure 3: Mean average success rate of algorithms for each task. Results are averaged within environment. Shaded areas represent ±1 std. over 5 seeds.

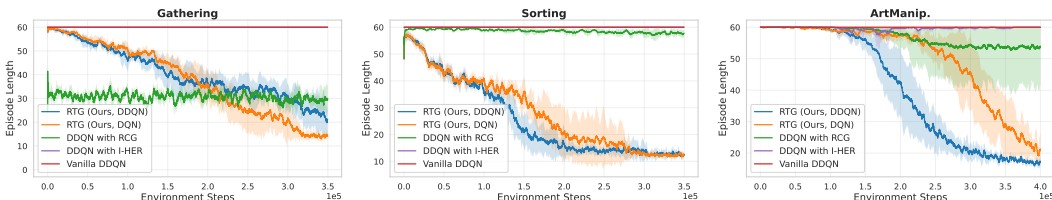

Figure 4: Mean episode length of algorithms for each task. Results are averaged within environment. Shaded areas represent ±1 std. over 5 seeds.

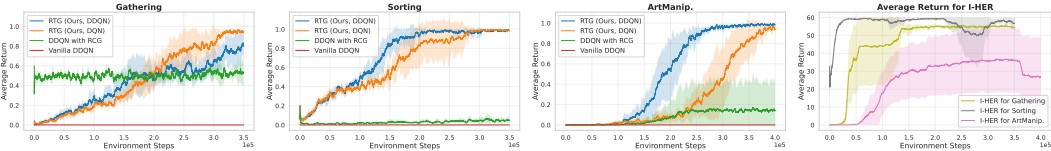

Figure 5: Mean average return of algorithms for each task. Results are averaged within each environment. Shaded areas represent ±1 std. over 5 seeds. I-HER does not terminate once reaching $r_{\text{HER}}(s_t, a_t) = 1$, but only upon reaching actual task success. Consequently, its episode return spans a larger range than other methods.

is capable of achieving high success rates on every task within a reasonable compute budget. We refer readers to Appendix C for visualizations of how each task is accomplished.

We ablate the design choice of how offline data is leveraged in our framework, as illustrated in Table 1. Our results indicate that, in contrast to Behavior Cloning (BC) or offline RL methods like Conservative Q-Learning (CQL) (Kumar et al., 2020), our method RTG incorporating weakly-guided offline transitions into the replay buffer of off-policy training leads to more effective learning. We refer readers to Appendix D for further ablations regarding the informativeness of offline transitions, and Appendix F for the Forward Replay gap analysis.

Table 1: Ablations on how to leverage offline transitions. For each task and algorithm, we report the percentage success rate (average and ±1 std. over 5 seeds).

| Task | BC | CQL | Ours |
|---|---|---|---|
| Gathering | 2.0 ± 2.4 | 2.0 ± 2.4 | **70.0 ± 21.6** |
| Sorting | 14.0 ± 7.3 | 16.0 ± 13.9 | **94.7 ± 4.6** |
| ArtManip. | 5.0 ± 5.4 | 8.0 ± 6.0 | **74.0 ± 21.9** |

We further evaluate the robustness and generality of our approach through four additional experiments conducted on the ArtManip task.

**Sim-to-sim transfer to Box2D.** We reused the reverse trajectories generated by our method as offline data for a standard Box2D implementation of the task, and trained a forward RL agent purely from these demonstrations. As shown in Figure 6, the agent achieves strong performance, indicating that our generated trajectories are physically consistent and transferable across simulators.

**Continuous control with Actor-Critic methods.** To evaluate generalization to continuous action spaces, we applied our method to a continuous variant of the ArtManip task, using Twin Delayed DDPG (TD3) (Fujimoto et al., 2018). The action space is parameterized as $(x, y, \Delta x, \Delta y)$, where $(x, y)$ specifies the pusher's planar position, and $(\Delta x, \Delta y)$ the pusher's displacement in the $x$- and $y$-directions. For the reverse step, we handle the continuous action space by discretizing it via sampling candidate actions. The results in Figure 7 show that our method continues to provide effective guidance in this continuous-control setting.

**Comparison with Backplay.** We further compared our method against Backplay (Resnick et al., 2018), a strong reset-to-state baseline that also exploits demonstrations. As shown in Figure 8, our method consistently outperforms Backplay, highlighting the advantage of optimizing full reverse trajectories rather than only replaying along a single forward demonstration.

**Beam search ablation.** Finally, we ablated the beam search depth D and breadth B used in Algorithm 1. Figure 9 shows that performance is robust across a wide range of beam breadths and

moderate changes in depth, suggesting that our method does not rely on exhaustive search to be effective. The runtime analysis can be found in Appendix E.

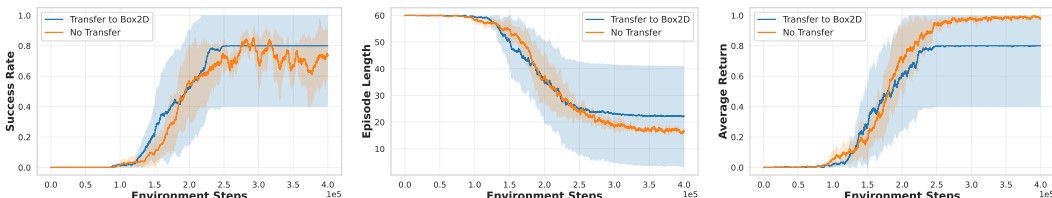

Figure 6: Sim-to-sim transfer to a standard Box2D implementation using our generated reverse trajectories with RTG. Results are averaged within environment. Shaded areas represent ±1 std. over 5 seeds.

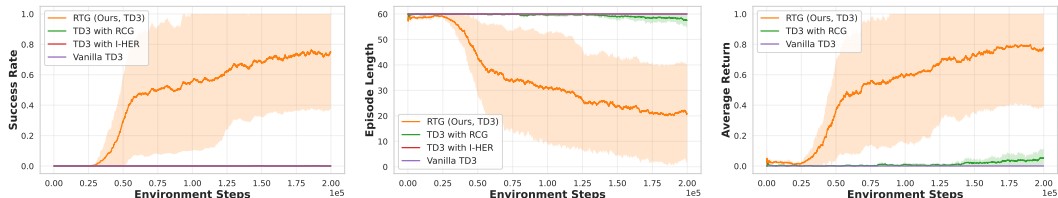

Figure 7: Continuous-control extension using TD3 as the underlying DRL algorithm, where the action space is $(x, y, \Delta x, \Delta y)$ and the reverse step operates on a sampled discretization of this space. Results are averaged within environment. Shaded areas represent ±1 std. over 5 seeds.

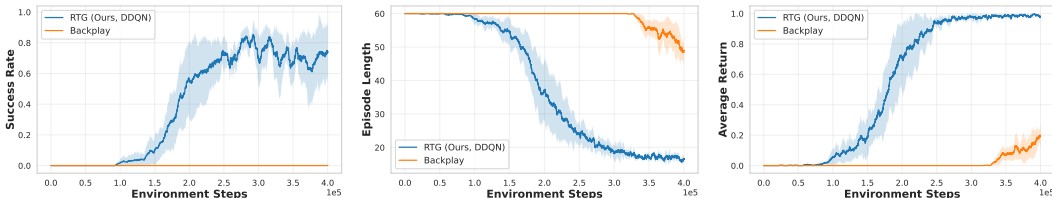

Figure 8: Comparison between our method and the Backplay baseline. Results are averaged within environment. Shaded areas represent ±1 std. over 5 seeds.

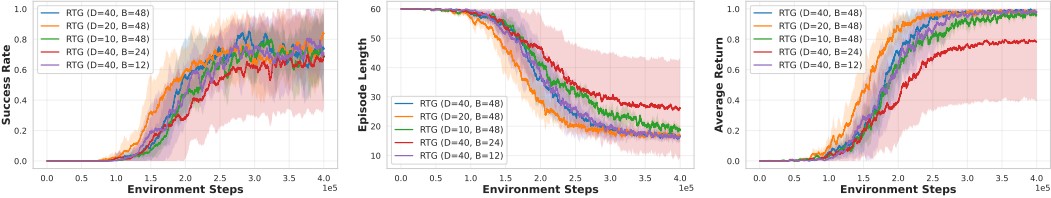

Figure 9: Ablation over beam search depth D and breadth B in Algorithm 1. Performance remains robust across various beam widths and depths, indicating that exhaustive search is not required. Results are averaged within environment. Shaded areas represent ±1 std. over 5 seeds.

## 6 CONCLUSION

In this work, we introduce RTG, a sample-efficient DRL method for learning rigid body manipulation skills under sparse reward. The core idea is to leverage trajectory optimization based simulator RRBS to generate reverse trajectories that terminate at high-reward states, and to employ beam search to construct a dataset $\tilde{\mathcal{D}}$ that augments the replay buffer of an off-policy DRL agent like DQN and DDQN. We evaluate RTG on various multi-object manipulation tasks, including sorting, gathering, and articulated object manipulation. Experiments show that RTG substantially improves off-policy DRL's performance, outperforming baselines including simplified-task HER and RCG.

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

## A    REVERSE SIMULATION VIA PHYSICS-CONSTRAINED OPTIMIZATION

To derive the reverse simulator, we need to first understand the mechanism of the forward simulation. The forward simulation is formulated as the following unconstrained optimization:

$$\arg\min_{p_{t+1}} \Psi(p_{t+1}, p_t, p_{t-1}, c_t), \tag{5}$$

where we denote by $p_t$ the kinematic state at the $t$th time instance, and $c_t$ is the control input at the $t$th time instance. In our setting of 2D rigid bodies, $p_t$ is the concatenation of $p_t^i$ with $p_t^i$ being the kinematic state of the $i$th rigid body, i.e. $p_t^i = (x_t^i, y_t^i, \theta_t^i)$ and the complete state $s_t$ used in DRL is a concatenation of kinematic state and velocity, i.e. $s_t = (p_t, p_{t-1})$, where velocity can be recovered from finite difference. The function $\Lambda$ in our main paper is defined as $\Lambda(s_{t+1}, s_t, c_t, q(a_t)) = \nabla_{p_{t+1}} \Psi(p_{t+1}, p_t, p_{t-1}, c_t)$. In Section A.1, we would introduce the formulation of the objective function $\Psi$. For now, we assume that $\Psi$ is twice-differentiable in all the parameters. In the reverse simulator, we consider an entire trajectory of $h$ timesteps, denoted as $p = (p_{t-h}^T, \cdots, p_{t-1}^T)^T$, with the associated control inputs $c = (c_{t-h+1}^T, \cdots, c_t^T)^T$, where we further denote by $p$ (resp. $c$) (without subscript) the concatenation of $p_t$ (resp. $c_t$) over all the time indices. Here, we assume $s_{t+1} = (p_{t+1}, p_t)$ is known as fixed. We would like to optimize the sequence of control inputs to optimize the following objective function:

$$J(p, c) = \sum_{i=1}^{M} (x_{t-h}^i, y_{t-h}^i) d(a_t) + \lambda \sum_{k=0}^{h-1} \|c_{t-k}\|^2 + P_\perp(p_{t-h}), \tag{6}$$

where our primary goal is to move all $M$ rigid bodies along the negative pushing direction $d(a_t)$ as far as possible, while fixing the final state. Note that our objective encourages the robot to push all $M$ rigid bodies, since the set of rigid bodies to be pushed simultaneously is unknown to us a prior. Note that when the robot cannot reach certain rigid bodies, these bodies will not move due to our physics constraints, despite our objective function encourages the bodies to be moved. Further, we also add a small control regularization with a small coefficient $\lambda$. Finally, we introduce a regularization energy $P_\perp(p_{t-h})$ to ensure the initial state satisfies the collision-free constraints, which is defined in Section A.1. During our optimization, we need to always ensure that Equation 5 is satisfied, which guarantees physical correctness. Combining Equation 5 and Equation 6, we propose to solve the following constrained optimization:

$$\arg\min_{p,c} J(p, c) \quad \text{s.t.} \nabla_{p_{t-k+1}} \Psi(p_{t-k+1}, p_{t-k}, p_{t-k-1}, c_{t-k}) = 0 \quad \forall k = 0, \cdots, h-1. \tag{7}$$

Under the assumption that the function $\Psi$ is twice-differentiable and thus the function $\Lambda$ is differentiable, we can efficiently solve Equation 7 using SQP.

### A.1    OPTIMIZATION-BASED 2D RIGID BODY SIMULATOR

In this section, we consider the dynamics of multiple 2D rigid bodies, for which we derive the concrete form of the objective $\Psi$ and its derivatives. Our starting point is the 2D version of the dynamic simulator (Huang et al., 2024). The energy $\Psi$ consists of five terms: the inertia term $I(p_{t+1}, p_t, p_{t-1})$ and damping term $I_D(p_{t+1}, p_t)$, the normal collision potential $P_\perp(p_{t+1})$, the frictional collision potential $P_\parallel(p_{t+1}, p_t)$, and finally the external force potential $P_E(p_{t+1}, c_t)$. Specifically, we have:

$$\Psi(p_{t+1}, p_t, p_{t-1}, c_t) = I(p_{t+1}, p_t, p_{t-1}) + I_D(p_{t+1}, p_t) + P_\perp(p_{t+1}) + P_\parallel(p_{t+1}, p_t) + P_E(p_{t+1}, c_t).$$

We present the concrete formula for each and every term above.

**Inertia & Damping Term:** In the original formula for the dynamic simulator (Huang et al., 2024), the inertial term is designed for soft bodies instead of rigid bodies. Instead, we follow Pan & Manocha (2018) to formulate the rigid body inertia term as follows:

$$I(p_{t+1}, p_t, p_{t-1}) = \sum_{j=1}^{M} \int_{\Omega_j} \frac{\rho}{2\Delta t^2} \|X(x, p_{t+1}^j) - 2X(x, p_t^j) + X(x, p_{t-1}^j)\|^2 dx, \tag{8}$$

where $\rho$ is the rigid body density, $\Delta t$ is the timestep size, and $\Omega_j \subset \mathbb{R}^2$ is the volume taken by the $j$th rigid body. Finally, $X(x, p_t^j)$ is the world-space position of $x$ under configuration $p_t^j$, defined as:

$$X(x, p_t^j) = \begin{pmatrix} \cos(\theta_t^j) & -\sin(\theta_t^j) \\ \sin(\theta_t^j) & \cos(\theta_t^j) \end{pmatrix} x + \begin{pmatrix} x_t^j \\ y_t^j \end{pmatrix}.$$

The dynamic simulator discretizes the acceleration by finite difference over three time instances. Following the similar logic to the inertial term, we can define the following damping term that penalizes the velocity at every timestep:

$$I_D(p_{t+1}, p_t) = k_D \sum_{j=1}^{M} \int_{\Omega_j} \frac{\rho}{2\Delta t^2} \|X(x, p_{t+1}^j) - X(x, p_t^j)\|^2 dx,$$

with $k_D$ being the damping coefficient. We refer readers to Pan & Manocha (2018) for the computational evaluation of these terms.

**Normal Collision Potential:** Without a loss of generality, we can assume the $i$th rigid body has the geometry of a convex polyhedron with $K_i$ vertices denoted as $(v_{i,1}, \cdots, v_{i,K_i})$. We now define the smoothened signed distance function of a point $p$ to the $i$th rigid body to be $d_i(p)$. We follow the method of incremental potential contact used by Huang et al. (2024) and define:

$$P_\perp(p_{t+1}) = -\nu \sum_{j=1}^{M} \sum_{i \neq j} \sum_{k=1}^{K_i} \log(d_j([v_{i,k}^j]_{t+1})),$$

where we choose $\nu$ as a small positive coefficient and $X^{-1}(\bullet, p_j^{t+1})$ is the inverse function of $X(\bullet, p^{t+1})$. Here we define $[v_{i,k}^j]_{t+1} = X^{-1}(X(v_{i,k}, p_i^{t+1}), p_j^{t+1})$, with is the position of $v_{i,k}$ in $j$th object's local frame of reference at time instance $t + 1$. In other words, $P_\perp$ requires that every vertex of a rigid body to be non-penetrating with other rigid bodies.

**Frictional Collision Potential:** The frictional potential is formulated in a similar manner following the idea of incremental potential contact used by Huang et al. (2024). We first compute each contact force between $v_{i,k}$ and the $j$th rigid body from the last timestep, which is:

$$f_{\perp,j,i,k} = \nu \left\| \frac{\partial \log(d_j([v_{i,k}^j]_t))}{\partial [v_{i,k}^j]_t} \right\|.$$

We then formulate the frictional damping term as:

$$f_{\parallel,j,i,k} = \beta f_{\perp,j,i,k} \left\| \text{Proj}_\parallel \left[ \frac{X(v_{i,k}, p_{t+1}^i) - X(v_{i,k}, p_t^i)}{\Delta t} - \frac{X([v_{i,k}^j]_t, p_{t+1}^j) - X([v_{i,k}^j]_t, p_t^j)}{\Delta t} \right] \right\|,$$

where $\text{Proj}_\parallel$ is the projection to the tangential plane. Finally, we define:

$$P_\parallel(p_{t+1}, p_t) = \nu \sum_{j=1}^{M} \sum_{i \neq j} \sum_{k=1}^{K_i} f_{\parallel,j,i,k},$$

with $\beta$ being the frictional coefficient. Intuitively, we damp the relative tangential velocity between $v_{i,k}$ on the $i$th object and $[v_{i,k}^j]_t$ on the $j$th object in contact.

**External Force Term:** We control the dynamic system using external force and torque. Without the loss of generality, we can assume the first rigid body is the robot end-effector, which can be controlled by $c_t = (f_t^x, f_t^y, \tau_t)$ with $(f_t^x, f_t^y)$ being the external force and $\tau_t$ being the external torque. Then the external force term is $-f_t^x x_t^1 - f_t^y y_t^1 - \tau_t \theta_t^1$. However, the above formula might introduce excessively large forces, which is unrealistic. We can regularize the situation by introduce a bound $B_f$ on the force magnitude and enforcing $-B_f \leq f_t^{x,y} \leq B_f$. Similarly, we introduce a bound $B_\tau$ on the torque magnitude and enforce $-B_\tau \leq \tau_t \leq B_\tau$. Such constraint can be achieved by using the $\tanh$ activation function and defining:

$$P_E(p_{t+1}, u_t) = -B_f \tanh(f_t^x) x_t^1 - B_f \tanh(f_t^y) y_t^1 - B_\tau \tanh(\tau_t) \theta_t^1.$$

### A.2 SEQUENTIAL QUADRATIC PROGRAMMING

We provide the complete detail of our SQP algorithm. We first define the constraint vector $C(p, c) = (\Lambda_{t-h+1}, \cdots, \Lambda_t)$, where we abuse notation and write $\Lambda_{t-k} = \Lambda(s_{t-k+1}, s_{t-k}, c_{t-k})$. We adopt the variant of SQP guided by the following $l_1$-merit function (Boggs & Tolle, 1995): $\Theta(p, c, \eta) = J(p, c) + \eta\|C\|_1$. We start from the initial guess $p^0 = (p_{t+1}, \cdots, p_{t+1}), c^0 = (0, \cdots, 0)$,

i.e., we initialize the trajectory to be static at state $p_{t+1}$ with all zero control forces and torques. SQP iteratively updates the solution $p, c$ to reduce the merit function until a critical point is achieved. To update a solution $p, c$, we solve the following quadratic programming by using quadratic approximation of the objective function and linear approximation of all the constraints:

$$\underset{\Delta p, \Delta c}{\arg\min} \frac{\partial J}{\partial p}^T \Delta p + \frac{\partial J}{\partial c}^T \Delta c + \frac{1}{2} \Delta p^T \frac{\partial^2 J}{\partial p^2} \Delta s + \frac{1}{2} \Delta c^T \frac{\partial^2 J}{\partial c^2} \Delta c$$

$$\text{s.t.} \quad C + \frac{\partial C}{\partial p} \Delta p + \frac{\partial C}{\partial c} \Delta c = 0. \tag{9}$$

The above Quadratic Program (QP) has all-linear equality constraints with a quadratic objective. This is because we use the $\tanh$ soft activation function to model control force and torque limits in the external force term $P_E$ from Section A.1, which transforms the inequality control limits into the equality constraints after linearization. This is key to the fast solution of trajectory optimization, since the QP sub-problem can be efficiently solved via the following KKT linear system:

$$\begin{pmatrix} \frac{\partial^2 J}{\partial p^2} & & \frac{\partial C}{\partial p}^T \\ & \frac{\partial^2 J}{\partial u^2} & \frac{\partial C}{\partial c}^T \\ \frac{\partial C}{\partial p} & \frac{\partial C}{\partial c} & \end{pmatrix} \begin{pmatrix} \Delta p \\ \Delta c \\ \lambda \end{pmatrix} = \begin{pmatrix} -\frac{\partial J}{\partial p} \\ -\frac{\partial J}{\partial c} \\ -C \end{pmatrix}, \tag{10}$$

with $\lambda$ being the associated Lagrangian multiplier. Note that by definition, the mixed derivatives $\frac{\partial^2 J}{\partial p^2} c$. For now, we assume the KKT-system can be readily solved, then SQP proceeds by choosing a step size $\eta^j$ such that:

$$\Theta(p + \eta^j \Delta p, c + \eta^j \Delta c, \eta) < \Theta(p, c, \eta) + \eta^j D\Theta(p, c, \Delta p, \Delta c, \eta), \tag{11}$$

where $D\Theta$ is the directional derivative of $\Theta$ along $\Delta p$ and $\Delta c$. To ensure that such $\eta^j$ exists, we need to choose $\eta > \|\lambda\|_\infty$. We notice that SQP is an infeasible solver that is not guaranteed to return a feasible solution. Specifically, a feasible solution is only returned when the constraint qualifications are satisfied. In practice, we find the constraint qualifications can be violated when the physics constraints are violated. To improve the success rate of SQP, we follow Solodov (2009) to use a feasibility safe-guard. Specifically, instead of using Equation 11 as the only condition of line-search, we add a condition to ensure that $\|C(p + \delta p, c + \Delta c)\|_1 \le \epsilon_c$. We find that by using a sufficiently small $\epsilon_c$, the SQP solver becomes much more robust and we never observe failure cases in our experiments.

## A.3 FAST KKT SYSTEM SOLVE

Directly solving Equation 10 without exploiting the sparsity pattern can take $O(h^3)$. Instead, we show that by utilizing the sparsity pattern, we can solve the KKT-system at a cost of $O(h)$. To see this, we write the Lagrangian multiplier $\lambda = (\lambda_{t-h+1}, \cdots, \lambda_t)$. The fast linear system solve can be derived by a permutation of variables as follows:

$$\nu_{t-k} \triangleq \begin{pmatrix} \Delta p_{t-k-1} \\ \Delta c_{t-k} \\ \lambda_{t-k} \end{pmatrix} \quad \begin{pmatrix} \Delta p \\ \Delta c \\ \lambda \end{pmatrix} = P \begin{pmatrix} \nu_{t-h+1} \\ \vdots \\ \nu_t \end{pmatrix},$$

with $P$ being the permutation matrix. The lefthand side of Equation 10 after symmetric permutation reads:

$$P \begin{pmatrix} \frac{\partial^2 J}{\partial p^2} & & \frac{\partial C}{\partial p}^T \\ & \frac{\partial^2 J}{\partial u^2} & \frac{\partial C}{\partial c}^T \\ \frac{\partial C}{\partial p} & \frac{\partial C}{\partial c} & \end{pmatrix} P^T = \begin{pmatrix} A_{t-h+1} & B_{t-h+2}^T & & & \\ B_{t-h+2} & A_{t-h+2} & B_{t-h+3}^T & & \\ & B_{t-h+3} & A_{t-h+3} & & \\ & & & \ddots & B_t^T \\ & & & B_t & A_t \end{pmatrix},$$

which takes a block tridiagonal form. Here we define the blocks as follows:

$$A_{t-k} \triangleq \begin{pmatrix} \frac{\partial^2 J}{\partial \Delta p_{t-k-1}^2} & & \frac{\partial \Lambda_{t-k}}{\partial \Delta p_{t-k-1}}^T \\ & \frac{\partial^2 J}{\partial \Delta c_{t-k}^2} & \frac{\partial \Lambda_{t-k}}{\partial \Delta c_{t-k}}^T \\ \frac{\partial \Lambda_{t-k}}{\partial \Delta p_{t-k-1}} & \frac{\partial \Lambda_{t-k}}{\partial \Delta c_{t-k}} & \end{pmatrix} \quad B_{t-k} \triangleq \begin{pmatrix} & & \frac{\partial \Lambda_{t-k-1}}{\partial \Delta p_{t-k-1}}^T \\ & & \\ & \frac{\partial \Lambda_{t-k}}{\partial \Delta p_{t-k-2}} & \end{pmatrix}$$

Therefore, we could use the cyclic reduction algorithm (Gander & Golub, 1998) to solve the linear system with a cost of $O(h)$. Put together, each iteration of our SQP involves a single solve of the KKT-system, so the iterative cost is $O(h)$.

## B   HYPERPARAMETERS

In Table 2, we report the choice of our method's hyperparameters.

Table 2: Our method's hyperparameters. These are the ones used to generate our figures and results. Highlighted in blue indicates hyperparameters introduced by this paper.

| Hyperparameter | Value |
| --- | --- |
| **RL Hyperparameters (DQN & Double DQN)** | |
| Discount factor ($\gamma$) | 0.8 (Gathering) |
| | 0.9 (Sorting and ArtManip.) |
| Replay Buffer Capacity | 1,500,000 |
| Batch Size | 512 |
| Total Interactions / Samples | 350,000 (Gathering and Sorting) |
| | 400,000 (ArtManip.) |
| **Networks and Optimization** | |
| Network Shape of Features Extractor (MLP) | [512, 512, 128] |
| Learning Rate | 5e-5 |
| Gradient Steps | 1 |
| Train Frequency | 4 |
| Network Optimizer | Adam |
| **Environment and Data** | |
| Reward Function | Sparse (+1 on success, 0 otherwise) |
| Action Repeat | 1 |
| Episode Horizon | 60 |
| Observation Type | State |
| **RTG** | |
| Push Stride | 2.0 |
| Number of discrete actions | 48 |
| Offline transitions generated | 49144 (Gathering) |
| | 67793 (Sorting) |
| | 89905 (ArtManip.) |

## C   TASK VISUALIZATIONS

In Figure 10, Figure 11, Figure 12, we show visually how each of our proposed task is accomplished.

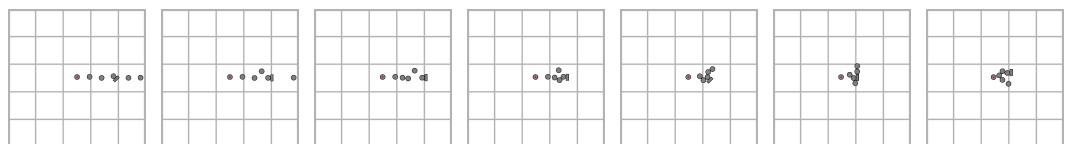

Figure 10: A sample successful trajectory for the task of Gathering.

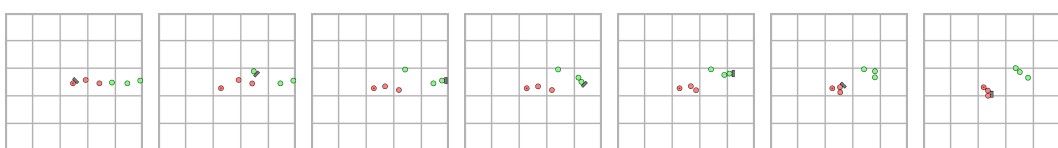

Figure 11: A sample successful trajectory for the task of Sorting.

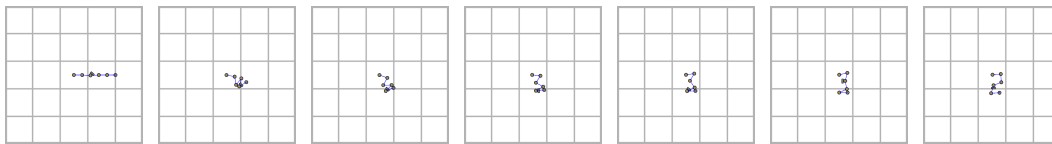

Figure 12: A sample successful trajectory for the task of ArtManip.

# D  ADDITIONAL RESULTS

Here we report more comparisons with (a) DDQN with Random Offline Exploration and (b) RTG with Domain Randomization, both combined with DDQN. For the former, DDQN with random offline exploration is essentially an ablation to validate the data informativeness of offline transitions from RTG. For the latter, we add domain randomization to the initial state distribution (with each object's positions perturbed by ±0.5, ±1.0, ±0.5 with respect to three tasks) to validate our method RTG's robustness. The comparisons are shown in Figure 13, Figure 14, Figure 15.

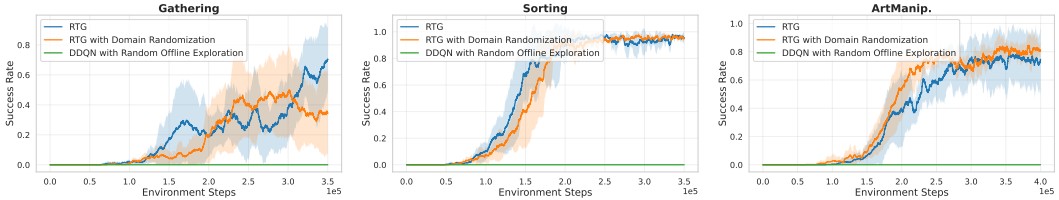

Figure 13: Mean average success rate of algorithms for each task. Results are averaged within each environment. Shaded areas represent ±1 std. over 5 seeds.

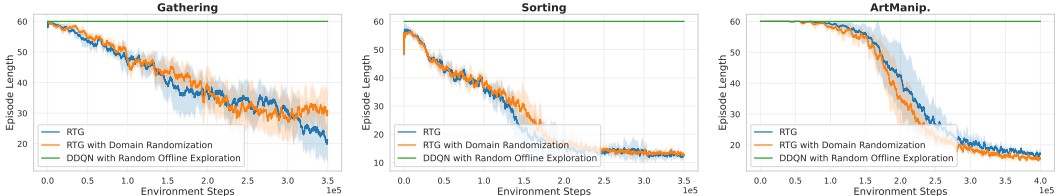

Figure 14: Mean episode length of algorithms for each task. Results are averaged within each environment. Shaded areas represent ±1 std. over 5 seeds.

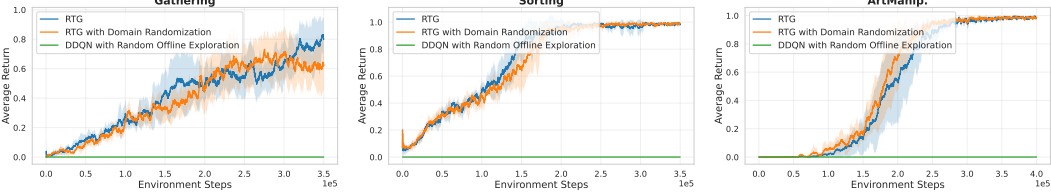

Figure 15: Mean average return of algorithms for each task. Results are averaged within each environment. Shaded areas represent ±1 std. over 5 seeds.

# E  RUNTIME ANALYSIS

**Runtime of backward and forward simulation.** During backward generation, we run our RRBS in quasi-static mode, with maximum solver iteration set to 1000. On an AMD Ryzen 9 5950X CPU (16C/32T, 1 socket, 1 NUMA node), a single parallel backward optimization over 48 candidate trajectories takes $2.45 \pm 0.13$ s wall-clock time (mean ± std) for our task of ArtManip (with joints). This corresponds to $51.0 \pm 2.6$ ms per backward action, where each optimizer solves one per-action

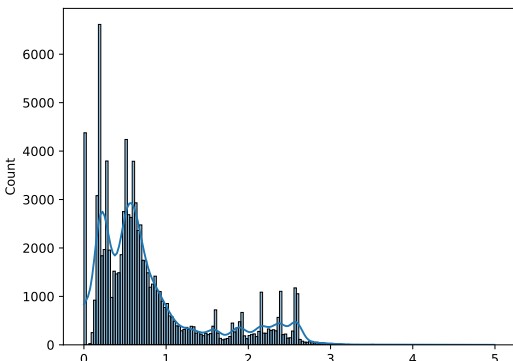

Figure 16: Replay gap distribution between reverse-generated states and the true forward dynamics. The horizontal axis shows the average 2D positional gap per node, measured as the mean Euclidean distance between each node's position in the backward-optimized state and the corresponding state after Forward Replay. Distances are in simulator units. The distribution is concentrated near zero but exhibits a non-zero tail, indicating a small yet systematic mismatch in reverse physics and thereby justifying our Forward Replay step.

backward simulation as defined in Figure 2(a). For Gathering and Sorting (without joints), the optimization stage over 48 candidates costs $1.90 \pm 0.06$ s, *i.e.*, $39.6 \pm 1.3$ ms per action step. For comparison, the forward simulator costs $1.07 \pm 0.34$ ms per action step, so a single-step backward optimization is approximately one order of magnitude more time-consuming than a forward step.

**Time complexity of beam search.** For our beam search (Algorithm 1) over backward actions with beam breadth $B$ and horizon depth $D$, at each search layer we expand at most $B$ nodes, and for each node we run one parallel backward optimization followed by a ranking step. Thus the total number of backward steps scales as $\mathcal{O}(D \times B)$, and the overall time complexity of the beam search is linear in both beam width and horizon depth.

## F  FORWARD REPLAY GAP ANALYSIS

We quantify the discrepancy between the states generated by our backward simulator and those produced by the true forward dynamics in Figure 16. As shown, the replay gap is small but clearly non-zero, indicating that the reverse physics are not perfectly consistent with the forward dynamics. This systematic mismatch motivates our Forward Replay step, which re-simulates backward-optimized trajectories under forward dynamics before utilizing them.

## G  USE OF LLMS

We acknowledge the use of large language models (LLMs) as assistive tools in this research. LLMs are used during paper writing, for improving grammar and wording. All outputs from these models were meticulously reviewed, revised, and verified by the authors, who retain full responsibility for all content presented in this paper.

