# OpenReview forum: "RTG: Reverse Trajectory Generation for Reinforcement Learning Under Sparse Reward"
_ICLR.cc/2026/Conference — Submitted to ICLR 2026_

### Official Review · Reviewer_M1gP · 2025-10-28

**Soundness:** 3
**Presentation:** 2
**Contribution:** 3
**Rating:** 6
**Confidence:** 4

**Summary:**

The authors propose Reverse Trajectory Generation (RTG) to address the sparse reward problem in the field of reinforcement learning (RL). Low sampling efficiency is indeed prevalent in complex RL tasks; to mitigate this, the authors introduce a reverse transition function to generate more beneficial training data, thereby accelerating the convergence of the RL policy. The authors’ idea is quite insightful, and their method achieves significant performance improvements compared to Hindsight Experience Replay (HER) and Reverse Curriculum Generation (RCG).

**Strengths:**

1.The proposed Reverse Trajectory Generation (RTG) method integrates the advantages of HER and RCG. It not only supports multi-objective goal setting but also avoids the complexity of curriculum design.

2.The authors elaborate on the specific implementation pathway of RTG: a Reverse Rigid-Body Simulator (RRBS) is used to construct an optimization problem for inverse solution, and methods such as forward replay are proposed to optimize the generated reverse data.

3.The proposed method demonstrates significantly better performance than HER and RCG across three tasks.

**Weaknesses:**

1.The authors' title is limited to RL, yet their proposed method relies on the Reverse Rigid-Body Simulator (RRBS) and manipulation tasks. The authors should clearly define the scope of the paper: if the method is only targeted at rigid-body manipulation tasks, this specificity should be reflected in the title; conversely, if a broader applicability is claimed, the authors must provide a more generalizable method along with corresponding validation.

2.The baselines selected by the authors are insufficient, as comparisons are only made with the classical HER and RCG. In fact, numerous variants of HER and RCG have been proposed in existing literature, and the authors should consider incorporating more recent and representative baselines to strengthen the comparative analysis.

3.The current simulation setup is overly idealized. More complex task configurations should be considered, such as validation with a physical robotic arm. Even a simple scenario tested on real hardware would significantly enhance the persuasiveness of the method’s evaluation.

**Questions:**

Similar to Weakness 3, Sim2Real transfer is a classic challenge in RL. I am curious whether the reversely synthesized data can enhance the performance of the method in real-world deployment. If the authors can provide validation for this aspect, I would consider increasing my review score.

---

> ### Author Response · Authors · 2025-11-24
> **Response to Reviewer M1gP**
>
> We thank the reviewer for the positive assessment and for recognizing our idea. We appreciate your constructive suggestions regarding the paper's scope and baselines.
>
> > **W1: Title Specificity and Scope**
>
> **A:** We agree with the reviewer’s observation that the current title suggests a broader RL scope than the specific rigid-body manipulation context presented. We have thus revised the title and will revise the title in the camera-ready version to explicitly reflect the focus on **rigid-body manipulation**. We further revised **Section 1** to make it clear that our method assumes the transition function is invertible.
>
> > **W2: Different Baselines**
>
> **A:** We agree with the reviewer that more recent baselines would strengthen the comparison. In response, we have added the following comparisons.
>
> **Backplay**: We implemented the "Backplay" algorithm [1], which relies on a fixed small set of demonstrations and resets states to those in the demonstrations to construct a curriculum. As shown in **Figure 8**, our method demonstrates superior performance against Backplay by actively generating curricula rather than relying solely on resetting to previously visited states. The average return and success rate of both methods are summarized below.
>
>
> | Method | Average Return | Success Rate |
> |------|-------------|------------|
> | Backplay | 0.19 | **0.2%** |
> | RTG (Ours, DDQN) | 0.98 |**74.0%** |
>
> **Twin Delayed DDPG (TD3)**: To demonstrate applicability beyond discrete settings, we compared our method with an actor-critic baseline, Twin Delayed DDPG (TD3) [2] in a continuous control setting, as shown in **Figure 7**. Our method successfully improves data efficiency in this regime. The average return and success rate of TD3 (with RCG) vs. our RTG are summarized below.
>
> | Method | Average Return | Success Rate |
> |------|-------------|------------|
> | TD3 (with RCG) | 0.05 | **0.0%** |
> | RTG (Ours, TD3) | 0.77 |**75.0%** |
>
> > **W3 and Q1: Sim-to-Real and Simulation Complexity**
>
> **A:** We acknowledge the importance of Sim-to-Real transfer. While a full hardware deployment is currently outside the scope of this algorithmic study, we have addressed your concern about the "idealized" nature of our simulation by conducting a "Sim-to-Sim" transfer experiment.
>
> Specifically, we generated reverse trajectories using our Reverse Rigid-Body Simulator (RRBS) and successfully used them to train an agent in the widely-used Box2D physics engine. As shown in **Figure 6**, our method does not overfit to a single idealized simulator. This "Sim-to-Sim" success is a strong indicator of the method's potential for Sim-to-Real transfer, as it shows robustness across different physics implementations. The average return and success rate of RTG (No Transfer) vs. RTG (Transfer to Box2D) are summarized below.
>
> | Method | Average Return | Success Rate |
> |------|-------------|------------|
> | No Transfer | 0.98 | **74.0%** |
> | Transfer to Box2D | 0.80 |**80.0%** |
>
> **[1]** Resnick, Cinjon, et al. "Backplay: man muss immer umkehren." arXiv preprint arXiv:1807.06919 (2018).
>
> **[2]** Fujimoto, Scott, Herke Hoof, and David Meger. "Addressing function approximation error in actor-critic methods." International conference on machine learning. PMLR, 2018.

---

### Official Review · Reviewer_oWM2 · 2025-10-31

**Soundness:** 3
**Presentation:** 3
**Contribution:** 2
**Rating:** 4
**Confidence:** 3

**Summary:**

The paper proposes a reverse-trajectory pipeline for sparse-reward manipulation: optimize short reverse-time trajectories from the goal, use a small beam search to expand candidates, then “forward replay” them to ensure feasibility and add them into the replay buffer alongside online data. Evaluated on several 2-D pushing/arrangement tasks with value-based agents, the method reports faster success than standard off-policy RL and simplified HER/RCG variants.

**Strengths:**

• Clear idea: generate from the goal backward, then verify forward and learn from those transitions.
• Nice engineering: reverse trajectory optimization plus a lightweight search is practical.
• Positioning: bridges goal-based data augmentation (HER-style) and reset/backplay ideas without heavy curriculum design, which are easy to follow.

**Weaknesses:**

• Scope is narrow. The method is shown only in 2-D with discrete pushing primitives. The proposed approach aims to be general, but the paper does not show 3-D or continuous control; the authors should add at least one 3-D or continuous-torque benchmark.
• Baselines feel light. The comparison emphasizes simplified HER/RCG. The proposed method looks strong, but the paper does not show standard UVFA-HER or backplay/reset-to-state; the authors should include those or justify why they are inapplicable.
• Compute transparency. Reverse optimization and beam search add overhead. The method claims efficiency, but the paper does not show wall-clock, solver iterations, or nodes-expanded; the authors should report runtime vs. success and sensitivity to beam width and horizon.
• Distribution shift. Mixing reverse and online data can bias Q estimates. The approach is plausible, but the paper does not show stability or calibration; the authors should sweep mixing ratios and add simple conservatism checks (e.g., CQL-style diagnostics).
• Generality beyond DQN/DDQN. The method is claimed to be broad, but the paper does not show actor-critic or model-based planners, such as MuZero.

**Questions:**

• How does success vary with beam width and horizon? Please show success vs. compute (nodes, solve time, iterations).
• How much do forward replays alter reverse-optimized states? Quantify feasibility gaps before/after replay and their effect on learning.

Another fundamental issue is: for some tasks, the entropy will increase along the traj but shrink by reversing the trajs. How it is possible to generate the reverse traj? Taking an example of unplug the socket, the reverse traj are much much complicated and hard than original one. And others like articulations. I am doubting the generalization of such a method on real problems.

---

> ### Author Response · Authors · 2025-11-24
> **Response to Reviewer oWM2 (1/2)**
>
> We thank the reviewer for the detailed and constructive feedback. We have addressed your concerns regarding scope, baselines, and experimental analysis below.
>
> > **W1 & W5: Extension to Continuous Control & Actor-Critic**
>
> **A:** We appreciate the suggestion to broaden the scope. As reflected in the revised paper, our setting is 2D rigid body manipulation. To demonstrate our method’s generality, we conducted a new experiment applying it to a continuous control task using Twin Delayed DDPG (TD3, an Actor-Critic algorithm) [1].
>
> This addresses both the concern about continuous control (W1) and the applicability to Actor-Critic architectures (W5). The results, added to **Figure 7**, demonstrate that our method successfully integrates with continuous action spaces and actor-critic methods, outperforming standard baselines. The average return and success rate of TD3 (with RCG) vs. our RTG are summarized below.
>
> | Method | Average Return | Success Rate |
> |------|-------------|------------|
> | TD3 (with RCG) | 0.05 | **0.0%** |
> | RTG (Ours) | 0.77 |**75.0%** |
>
> We acknowledge that Model-Based RL (e.g., MuZero) is an interesting direction, but as our focus is on model-free sample efficiency, we believe the actor-critic TD3 experiment best demonstrates the method's generality within the paper's scope.
>
> > **W2: Standard Backplay/Reset-to-state Baseline**
>
> **A:** We would like to submit that our I-HER is a stronger version of UVFA-HER. This is because I-HER incurs a denser reward than UVFA-HER. Indeed, UVFA-HER is using a multi-goal setting, where the reward is only non-zero when one specified goal is achieved. Instead, I-HER would grant a non-zero reward when any of the goals is reached. Since I-HER does not perform well on our tasks, there is no need to further compare with UVFA-HER.
>
> We have also included a comparison with the **standard RCG** (which is a reset-to-state method) in our Figures 3,4,5,7. Finally, we have conducted a new experiment comparing our method against an additional baseline, **Backplay**, following the implementation in [2].
> Our revised **Figure 8** shows that our method achieves higher performance and data efficiency than Backplay. The average return and success rate of both methods are summarized below.
>
> | Method | Average Return | Success Rate |
> |------|-------------|------------|
> | Backplay | 0.19 | **0.2%** |
> | RTG (Ours, DDQN) | 0.98 |**74.0%** |
>
>
> > **W3 and Q1: Runtime and Beam Search Analysis**
>
> **A:** We agree that "compute transparency" is important. We have added a runtime analysis in **Appendix E** and an ablation study with beam width and horizon in **Figure 9**.
>
> **Runtime**: We report the wall-clock time in the revised **Appendix E**. The maximum solver iteration for our RRBS is set to 1000. On an AMD Ryzen 9 5950X CPU (16C/32T, 1 socket, 1 NUMA node), a single parallel backward optimization over 48 candidate trajectories takes $2.45 \pm 0.13$ s wall-clock time (mean $\pm$ std) for our task of ArtManip (with joints). This corresponds to $51.0 \pm 2.6$ ms per backward action, where each optimizer solves one per-action backward simulation as defined in Figure 2(a). For Gathering and Sorting (without joints), the optimization stage over 48 candidates costs $1.90 \pm 0.06$ s, i.e., $39.6 \pm 1.3$ ms per action step. For comparison, the forward simulator costs $1.07 \pm 0.34$ ms per action step, so a single-step backward optimization is approximately one order of magnitude more time-consuming than a forward step. While reverse generation adds overhead, the significant reduction in total environment interactions required for convergence results in a net improvement to training efficiency.
>
> **Beam Search Sensitivity**: We ablated beam width and depth in the revised **Figure 9**. The results indicate that performance is robust to these hyperparameters, and our method does not rely on exhaustive search to be effective. The average return and success rate for five variants are summarized below.
>
> | Method | Average Return | Success Rate |
> |------|-------------|------------|
> | RTG (D=10, B=48) | 0.96 ± 0.01 | **68.5 ± 26.7%** |
> | RTG (D=20, B=48) | 0.98 ± 0.02 |**83.6 ± 12.3%** |
> | RTG (D=40, B=12) | 0.98 ± 0.02 |**69.8 ± 25.3%** |
> | RTG (D=40, B=24) | 0.79 ± 0.39 |**69.4 ± 34.8%** |
> | RTG (D=40, B=48) | 0.98 ± 0.01 |**74.0 ± 21.9%** |

---

> ### Author Response · Authors · 2025-11-24
> **Response to Reviewer oWM2 (2/2)**
>
> > **W4: Distribution Shift and Mixing Ratios**
>
> **A:** Regarding the concern about distribution shift when mixing online and offline reverse data, our design follows the setting studied in RLPD [3], which explicitly analyzes online off-policy RL with an auxiliary offline replay buffer. RLPD proposes a "symmetric sampling" strategy that draws 50% of each minibatch from online experience and 50% from the offline buffer, and shows through ablations across various benchmarks that performance is largely insensitive to the exact mixing proportion. A 50/50 ratio offers a good trade-off between sample efficiency, variance, and asymptotic return. In addition, RLPD demonstrates that this balanced sampling is able to keep Q-values bounded and achieve stable learning. Motivated by these findings, we adopt the same fixed mixing ratio. Empirically, we do not observe exploding Q-values or training instabilities, suggesting that explicit conservative penalties (e.g., CQL-style regularizers) are not necessary in our setting.
>
> > **Q2: Forward Replay Gap**
>
> **A:** We analyzed the discrepancy between the generated reverse states and the actual forward dynamics in our revised **Appendix F**. The average per-node distance between each replayed state is summarized below.
>
> | Statistic        | Distance  |
> |------------------|--------|
> | Mean             | 0.798  |
> | Std. deviation   | 0.699  |
> | Min              | 0.000  |
> | Max              | 4.993  |
> | Median           | 0.592  |
> | 25th percentile  | 0.289  |
> | 75th percentile  | 0.950  |
>
> Distances are in simulator units. For reference, the pusher has a size of $2.5 \times 1$.
> As demonstrated, the forward replay gap is small but clearly non-zero, indicating that the reverse physics are not perfectly consistent with the forward dynamics. This systematic mismatch validates the necessity of our Forward Replay step. By replaying the trajectory forward, we filter out infeasible transitions and ensure the replay buffer only contains valid physics data, effectively bridging the gap before the RL agent learns from it.
>
> > **Q3: Entropy and Reversibility of the Task**
>
> **A:** This is indeed a profound question. We agree with the reviewer that the forward entropy increases while the reverse entropy shrinks (many states map to one goal). However, we emphasize that we have already provided the mechanism to address this issue in our method (**Section 4.1**).
>
> Note that our method formulates the RTG as a trajectory optimization problem with a specified objective function (**Equation 3**). This implies that our method introduces bias in the "many states" towards the one state that minimizes the objective. This strategy is not only used by beam search to bias towards a given start configuration, but also injects information into the system dynamics to break symmetry and increase entropy when reversing the trajectories.
>
> **[1]** Fujimoto, Scott, Herke Hoof, and David Meger. "Addressing function approximation error in actor-critic methods." International conference on machine learning. PMLR, 2018.
>
> **[2]** Resnick, Cinjon, et al. "Backplay: man muss immer umkehren." arXiv preprint arXiv:1807.06919 (2018).
>
> **[3]** Ball, Philip J., et al. "Efficient online reinforcement learning with offline data." International Conference on Machine Learning. PMLR, 2023.

---

### Official Review · Reviewer_uWfL · 2025-10-31

**Soundness:** 3
**Presentation:** 2
**Contribution:** 2
**Rating:** 4
**Confidence:** 3

**Summary:**

The paper introduces two methods: 1) Reverse Rigid-Body Simulator (RRBS), and approach which leverage differentiability to generate short-range inverse models and b) Reverse Trajectory Generation (RTG), which uses RRBS to provide synthetic trajectories which reach desired goals which are then used for off-policy RL. Conceptually, the approach blends hindsight experience replay with reverse curriculum generation on the policy optimization side, while using differentiable inverse models to avoid reasoning over long sequences of forwards actions. The approach is demonstrated on several 2-D table top manipulation tasks.

**Strengths:**

$\textbf{Elegant approach}$: The approach is elegant, simple, and well motivated. The approach for exploring in trajectory space rather than forward action space is rather intriguing, and potentially a powerful general insight.

$\textbf{Clarity}$: The paper is well written and easy to understand. The paper effectively uses concrete examples to illustrate the main ideas behind the paper.

$\textbf{Performance on benchmarks}$: On the benchmarks presented, the paper demonstrates a clear performance boost over baselines.

**Weaknesses:**

$\textbf{Scalability}$: The simple 2-D test domains presented in the paper do not effectively demonstrate whether the approach is competitive with baselines in more realistic, high dimensional settings. Despite the elegance of the approach, the results are substantially below the bar for acceptance at ICLR. I can’t argue for acceptance without seeing improvements over baselines on more standard robotics benchmarks.

$\textbf{Specificity of solver}$: Even though the high-level idea is quite general, many of the details of the approach are tied to the specific physics solver used in the paper. Thus, it seems difficult for the community easily pick up the algorithm and use it off the shelf in different settings.  I strongly suggest the authors rework the algorithm in a more general, widely used differentiable physics engine.

**Questions:**

- Can the methods scale to standard high-dimensional robotics baselines?

- Is the differentiable inverse dynamics approach easily adaptable to other simulators?

- Why beam search verse other search heuristics? How would beam search scale to more high dimensional spaces?

---

> ### Author Response · Authors · 2025-11-24
> **Response to Reviewer uWfL**
>
> We thank the reviewer for the detailed evaluation and for recognizing our contribution. We appreciate your feedback regarding the experimental settings and have conducted additional experiments to address your concerns about scalability and generalization.
>
> > **W1 & Q1: Scalability to Different Settings**
>
> **A:** We acknowledge the reviewer's concern regarding the complexity of the test domains. To demonstrate that our method is not limited to discrete settings, we have conducted a new experiment on a continuous control task.
>
> In this experiment, we adapted RTG to handle continuous action spaces in the Articulated Manipulation (ArtManip) task and achieved higher success rates than baselines. The results are added to **Figure 7** of the revised PDF. The average return and success rate of TD3 (with RCG) vs. our RTG are summarized below.
> | Method | Average Return | Success Rate |
> |------|-------------|------------|
> | TD3 (with RCG) | 0.05 | **0.0%** |
> | RTG (Ours, Continuous) | 0.77 |**75.0%** |
>
> > **Q1: Scalability to Higher Dimension**
>
> We respectfully submit that the primary contribution of this work is the novel algorithmic framework (RTG) for solving sparse reward problems via reverse dynamics. The 2D manipulation tasks, now including continuous control, were chosen to isolate and analyze these algorithmic properties clearly. A 3D extension would require a complete reimplementation of our simulator, which is beyond the scope of this project. That said, we argue that our Reverse Rigid-Body Simulator (RRBS) is a 2D variant of [1], which is dimension independent, so a 3D extension is theoretically straightforward.
>
> > **W2 & Q2: Specificity of Solver and Generalization**
>
> **A:** This is closely related to the previous question. We agree with the reviewer that adopting our method would require generalization to other simulator formulations. To demonstrate that our method is not strictly tied to our specific physics solver, we conducted a new "sim-to-sim" experiment using the popular Box2D [2] engine.
>
> Specifically, we utilized the reverse trajectories generated by our simulator RRBS and successfully used them as demonstrations for off-policy learning in the standard Box2D environment via Forward Replay (**Section 4.2**). This confirms that the generated trajectories are physically consistent enough to be useful in widely used simulators like Box2D. We have added these results to **Figure 6** to show that the community can leverage our approach with standard tools. The average return and success rate of RTG (No Transfer) vs. RTG (Transfer to Box2D) are summarized below.
>
> | Method | Average Return | Success Rate |
> |------|-------------|------------|
> | No Transfer | 0.98 | **74.0%** |
> | Transfer to Box2D | 0.80 |**80.0%** |
>
> > **Q3: Search Heuristics**
>
> **A:** We selected Beam Search primarily for its simplicity and effectiveness in balancing exploration and exploitation during trajectory generation. However, a key strength of our proposed RRBS framework is that it is **agnostic to the search heuristic**; it can be combined with other search methods depending on the specific constraints of the domain.
>
> To address your question on scaling and sensitivity, we included an ablation study on beam width and depth in our revised **Figure 9**. It shows that our method’s performance is robust across a range of beam widths and depths, suggesting RTG does not require exhaustive search to be effective. The average return and success rate for five variants are summarized below.
>
> | Method | Average Return | Success Rate |
> |------|-------------|------------|
> | RTG (D=10, B=48) | 0.96 ± 0.01 | **68.5 ± 26.7%** |
> | RTG (D=20, B=48) | 0.98 ± 0.02 |**83.6 ± 12.3%** |
> | RTG (D=40, B=12) | 0.98 ± 0.02 |**69.8 ± 25.3%** |
> | RTG (D=40, B=24) | 0.79 ± 0.39 |**69.4 ± 34.8%** |
> | RTG (D=40, B=48) | 0.98 ± 0.01 |**74.0 ± 21.9%** |
>
> **[1]** Huang, Zizhou, et al. "Differentiable solver for time-dependent deformation problems with contact." ACM Transactions on Graphics 43.3 (2024): 1-30.
>
> **[2]** https://github.com/erincatto/box2d

---

### Official Review · Reviewer_KtVV · 2025-11-01

**Soundness:** 2
**Presentation:** 3
**Contribution:** 3
**Rating:** 6
**Confidence:** 3

**Summary:**

Exploration remains a key challenge when using reinforcement learning (RL) for sparse reward tasks, and many research papers have sought to address this problem from different angles: off-policy learning, goal relabeling (hindsight experience replay; HER), demonstrations, skill priors, as well as techniques that create training curricula by directly manipulating the initial state distribution of e.g. a simulation environment (reverse curriculum generation; RCG). This paper identifies key limitations of HER and RCG (limited use cases and reliance on human-designed curricula, respectively), and proposes reverse trajectory generation (RTG), a method that addresses these limitations by instead generating goal-reaching trajectories via optimization over a reversed transition function that models reverse-time environment dynamics (this work relies on a reverse rigid-body simulator in practice). Experiments are conducted on three simple 2D manipulation tasks with discrete actions, and empirical results indicate that the proposed method is significantly more data-efficiency than RCG and a slightly improved version of HER that the authors denote as implicit HER.

**Strengths:**

My initial assessment of this paper leans positive overall; it is very well written and technical contributions appear original and sound Specifically:
- I believe that this paper studies a relevant and timely problem (solving sparse reward RL problems in simulation), and is likely to be of interest to the community. The problem is clearly motivated, and shortcomings of existing work (HER and RCG in particular) are described in the introduction + related work sections. The paper is very well written, self-contained, and easy to follow; I believe that readers will be able to appreciate the technical contributions without much background knowledge. The illustration in Figure 1 is helpful for understanding the limitations of prior work.
- The proposed method seems technically sound and appears to rely fairly little on engineering or manual labor (compared to RCG) which I definitely consider a plus. It does rely on the assumption of a reliable (to the extent that the forward replay trick will work at least some of the time) reverse transition function, which I believe is not currently available in most robotics simulators, but it is interesting to see what is possible when you do have such privilege.
- The experiments seem well thought out and successfully demonstrate the advantage of the proposed method vs. HER and RCG on simple 2D manipulation tasks. The baselines seem reasonable.

**Weaknesses:**

While my overall assessment leans positive, I do have some reservations that I would like the authors to address:
- It is not clear to me how this method would be extended to continuous control tasks and/or discrete control tasks with a larger number of actions. Since the paper chooses to explicitly focus on robotics as application area (for which simulation is often used) which typically has a continuous action space and no reverse transition function readily available, I do believe that at the very least an attempt should be made at bridging this gap. For example, maybe an approximate reverse transition function could be obtained via sampling, and/or additional experiments on a simple 2D physics problem with continuous actions could be added.
- I am a bit skeptical about the BC and CQL results; can the authors please explain why there is such a big performance gap between these methods and the proposed method? If these methods have access to offline data that sufficiently cover the initial state distribution used during evaluation then I would expect it to be rather successful.
- This is relatively minor, but I would appreciate it if the authors can make the font size larger in their experimental result figures on page 9; I had to zoom in quite a bit in order to read the legend and axis labels.

**Questions:**

I would really appreciate it if the authors can address my comments in the "weaknesses" section above using written arguments and potentially additional experimental results. My main concerns / questions pertain to limitations, experimental setup, and baseline results.

---

> ### Author Response · Authors · 2025-11-24
> **Response to Reviewer KtVV**
>
> We thank the reviewer for the positive assessment and the constructive feedback. We appreciate your acknowledgment of the method's potential to reduce manual engineering in curriculum generation. We have addressed your specific concerns below.
>
> > **W1: Extension to Continuous Control**
>
> **A:** We appreciate this insightful suggestion regarding the applicability of our method to continuous control tasks. As suggested, we have conducted a new experiment to bridge this gap. Specifically, we applied our method to a continuous version of the Articulated Manipulation (ArtManip) task, using Twin Delayed DDPG (TD3) [1] as the underlying DRL algorithm. The action space is parameterized as $(x,y,\Delta x,\Delta y)$, where $(x,y)$ specifies the pusher’s planar position, and $(\Delta x,\Delta y)$ the pusher’s displacement in the $x$- and $y$-directions.
>
> The results, which we have added to our revised **Figure 7**, demonstrate that our method maintains its data efficiency advantage in this setting compared to the baselines. This confirms that the core logic of Reverse Trajectory Generation (RTG) is not limited to discrete spaces. The average return and success rate of TD3 (with RCG) vs. our RTG are summarized below.
>
> | Method | Average Return | Success Rate |
> |------|-------------|------------|
> | TD3 (with RCG) | 0.05 | **0.0%** |
> | RTG (Ours) | 0.77 |**75.0%** |
>
> > **W2: Why BC and CQL has a large performance gap**
>
> **A:** The performance gap stems from the inherent scarcity of the offline data available to these baselines. While the offline dataset contains successful goal-reaching trajectories from beam search, it does not sufficiently cover the broader state distribution encountered during evaluation and is essentially suboptimal. As a result, offline methods are forced to either imitate this behavior (BC) or remain conservative to avoid extrapolation error (CQL), preventing them from improving beyond the dataset. In contrast, our choice of utilizing such offline data with off-policy RL methods allows the agent to interact with the environment, refine its value estimates through trial-and-error, and explore actions not present in the offline dataset, thus enabling it to surpass the performance of purely offline RL methods.
>
> > **W3: Font Size**
>
> **A:** We apologize for the readability issues in the original submission. We have increased the font size for the new figures included in the rebuttal. We will update all figures to a larger, legible font size in the final camera-ready version to ensure clarity.
>
> **[1]** Fujimoto, Scott, Herke Hoof, and David Meger. "Addressing function approximation error in actor-critic methods." International conference on machine learning. PMLR, 2018.

---

> > ### Comment · Reviewer_KtVV · 2025-11-24
> > **Thank you**
> >
> > Thank you for your response. I appreciate the addition of a continuous control problem, as well as experiments and changes to writing that address concerns raised by other reviewers. I am willing to raise my score to 8 and recommend acceptance.

---

> > > ### Author Response · Authors · 2025-11-25
> > > **Response to Reviewer KtVV**
> > >
> > > Dear Reviewer KtVV,
> > >
> > > We are glad to hear that our rebuttal addressed your concerns well! Also, we appreciate your support for our work. If you have any further questions or suggestions, please do not hesitate to let us know.
> > >
> > >
> > > Best regards,
> > >
> > > Authors

---

### Author Response · Authors · 2025-11-24
**Common Response**

Dear reviewers and AC,

We sincerely appreciate your valuable time and effort spent reviewing our manuscript.

**Reiteration of Main Contribution**

We propose RTG (Reverse Trajectory Generation), a sample-efficient off-policy DRL method with RRBS (Reverse Rigid-Body Simulator) to generate trajectories that terminate at high-reward states to address the sparse reward problem in rigid-body manipulation.

We appreciate your constructive feedback on our manuscript. We are encouraged that the reviewers unanimously recognized the originality and potential of our approach.
Specifically, reviewers highlighted the method as **'elegant, simple, and well motivated'** (Reviewer `uWfL`), **'insightful'** (Reviewer `M1gP`), and **'technically sound'** (Reviewer `KtVV`). The committee also commended the **'nice engineering'** (Reviewer `oWM2`) and the **'clear performance boost'** (Reviewers `uWfL, M1gP`) our method achieves over standard baselines.


**Additional Experiments and Revisions**

The reviewers raised several questions regarding the method's generalization (to continuous control and other simulators), baselines (Backplay and actor-critic methods), and computational details (runtime, ablation, and physics consistency). In response to the comments, we have carefully revised and improved the manuscript as follows:

1. Generalization to Continuous Control (Reviewers `KtVV, uWfL, oWM2`): To address concerns about the discrete action space, we extended our framework to a continuous control task. We integrated our method with an actor-critic method TD3 (Twin Delayed DDPG) [1] and evaluated it on a continuous version of the ArtManip task. The revised **Figure 7** shows that our method significantly outperforms baselines in the continuous setting, demonstrating that our RTG is effective beyond discrete action spaces.
2. Generalization to General Physics Engines (Reviewers `uWfL, M1gP`): To demonstrate that our method is not limited to our specific solver, we conducted a Sim-to-Sim transfer experiment using the widely used Box2D [2] engine. We successfully used trajectories generated by our Reverse Rigid-Body Simulator (RRBS) to guide learning in a standard Box2D environment. The revised **Figure 6** confirms the physical consistency of our generated data and its transferability to standard tools.
3. Clarification of Scope (Reviewers `M1gP, oWM2`): We have revised the paper (and will update the title in the camera-ready version) to explicitly scope our contribution to Rigid-Body Manipulation. We clarified the assumptions regarding the invertibility of contact mechanics to ensure the claims are precise.
4. New Baseline of Backplay (Reviewers `oWM2, M1gP`): We added a comparison against Backplay [3], which uses a single demonstration to construct a curriculum for a given task. The revised **Figure 8** shows that our method outperforms Backplay, validating the advantage of generating reverse curricula via optimization rather than solely relying on replaying experienced states.
5. In-depth Algorithmic Analysis (Reviewers `uWfL, oWM2`): We have added additional analysis to improve transparency:
    - Beam Search Ablation: We analyzed the method's sensitivity to beam width and depth. Results in the revised **Figure 9** show the method is robust to these hyperparameters.
    - Compute Cost: In revised **Appendix E**, we provided a breakdown of wall-clock time for our forward and backward solvers, together with time complexity analysis for our beam search. The analysis shows that the overhead of reverse generation is outweighed by the significant gain in sample efficiency.
    - Forward Replay Gap: In revised **Appendix F**, we quantitatively analyzed the distribution shift between the generated reverse trajectories and the forward dynamics. This analysis validates the necessity of our Forward Replay mechanism.

In the revised manuscript, these updates are temporarily highlighted in blue for your convenience to check.

We sincerely believe that these updates may help us better deliver the benefits of the proposed RTG to the ICLR community. We look forward to further discussion.

Thank you very much,

Authors.





**[1]** Fujimoto, Scott, Herke Hoof, and David Meger. "Addressing function approximation error in actor-critic methods." International conference on machine learning. PMLR, 2018.

**[2]** https://github.com/erincatto/box2d

**[3]** Resnick, Cinjon, et al. "Backplay: man muss immer umkehren." arXiv preprint arXiv:1807.06919 (2018).

---

### Author Response · Authors · 2025-12-04
**Post Rebuttal Summary**

Dear ACs, PCs, and reviewers,

Thanks for all of your efforts in this period. To save your time, we summarize the status of the rebuttal discussion below:

1. Our paper proposes RTG (Reverse Trajectory Generation), a sample-efficient off-policy DRL method with RRBS (Reverse Rigid-Body Simulator) to generate trajectories that terminate at high-reward states to address the sparse reward problem in rigid-body manipulation.
2. We appreciate the reviewers' unanimous recognition of the originality and potential of our approach, including 'elegant, simple, and well motivated' (Reviewer `uWfL`), 'insightful' (Reviewer `M1gP`), 'technically sound' (Reviewer `KtVV`), 'nice engineering' (Reviewer `oWM2`) and the 'clear performance boost' (Reviewers `uWfL, M1gP`).
3. We have addressed all major concerns raised by the reviewers, including Generalization to Continuous Control, Generalization to General Physics Engines, Clarification of Scope, New Baseline of Backplay, and In-depth Algorithmic Analysis. For details, please refer to our **Common Response**.
4. We thank Reviewer `KtVV` for the prompt reply to our rebuttal, raising the score from 6 to 8. We note that this update was made days before the OpenReview API security incident.

Finally, we express our gratitude once again for your contributions to this conference.


Sincerely,

Authors

---

### Meta-Review · Area_Chair_LHpJ · 2026-01-03

**Summary:**

This is a boarderline paper. While Reviewer KtVV changed their score in favor of acceptance, many reviewers (uWfL, oWM2, M1gP) raise concern about the limited applicability of the proposed method beyond simple domain. This concern comes from the need to being able to perform trajectory optimization using an reversible model, which is non trivial to obtain generally, as the authors also mentioned in their rebuttal "A 3D extension would require a complete reimplementation of our simulator, which is beyond the scope of this project.". In addition, despite the authors adding a new task of continuous actions, it is still limited to a 2D environment, and the sim-to-sim transfer problem is still toy from a manipulation perspective. Therefore, given the current results, it is unclear whether the proposed method can be easily adopted for problems outside what were demonstrated in the current experiments. I would encourage the authors to present also results on realistic 3D manipulation tasks and preferably on real robots as well.

Minor: Reviewers also raised the point of computation overhead. While the authors show wall time, I think this is not sufficient. The plots in the current experiments show an x-axis of Environment Steps. I think this should account for the number of reverse environment steps used by the trajectory optimizer as well for a fair comparison, unless the experiments is about transfer to a different environment whose dynamics are different.

**Reviewer Concerns:**

Most of the technical questions are addressed. I think the limited scope and applicability issue remains.

**Reviewer Scores:**

I think the reviewers (except KtVV) would likely not change the score here, as the reviewers do not seem super excited and the fundamental narrow scope issue is not addressed sufficiently in the rebuttal in my opinion.

---

### Decision · Program_Chairs · 2026-01-26

Reject